# Polysaccharides induce deep-sea *Lentisphaerae* strains to release chronic bacteriophages

Chong Wang[1,2,3], Rikuan Zheng[1,2,3], Tianhang Zhang[1,2,3,4], Chaomin Sun[1,2,3,4]*

[1]CAS and Shandong Province Key Laboratory of Experimental Marine Biology and Center of Deep Sea Research, Institute of Oceanology, Chinese Academy of Sciences, Qingdao, China; [2]Laboratory for Marine Biology and Biotechnology, Qingdao Marine Science and Technology Center, Qingdao, China; [3]Center of Ocean Mega-Science, Chinese Academy of Sciences, Qingdao, China; [4]College of Earth Science, University of Chinese Academy of Sciences, Beijing, China

**Abstract** Viruses are ubiquitous in nature and play key roles in various ecosystems. Notably, some viruses (e.g. bacteriophage) exhibit alternative life cycles, such as chronic infections without cell lysis. However, the impact of chronic infections and their interactions with the host organisms remains largely unknown. Here, we found for the first time that polysaccharides induced the production of multiple temperate phages infecting two deep-sea *Lentisphaerae* strains (WC36 and zth2). Through physiological assays, genomic analysis, and transcriptomics assays, we found these bacteriophages were released via a chronic style without host cell lysis, which might reprogram host polysaccharide metabolism through the potential auxiliary metabolic genes. The findings presented here, together with recent discoveries made on the reprogramming of host energy-generating metabolisms by chronic bacteriophages, shed light on the poorly explored marine virus–host interaction and bring us closer to understanding the potential role of chronic viruses in marine ecosystems.

*For correspondence:
sunchaomin@qdio.ac.cn

Competing interest: The authors declare that no competing interests exist.

## eLife assessment

This manuscript presents **valuable** findings on two isolates of deep sea Lentisphaerae strains, which further our understanding of deep sea microbial life. The manuscript's primary claim is that phage isolates augment polysaccharide use in Pseudomonas bacteria, with preliminary evidence for the potential auxiliary metabolic genes in chronic phage infection and/or host proliferation. The strength of the evidence is overall **solid** and there are only minor weaknesses regarding the mechanism of polysaccharide use by the phages and the evidence for chronic infection. Overall, the data on Lentisphaerae strains will deepen our understanding of microbial life in the deep sea.

## Introduction

Viruses are the most abundant and genetically diverse biological entities on this planet (*Suttle, 2007*), and potentially play a role in shaping microbial abundances, community structure, and the evolutionary trajectory of core metabolisms (*Sullivan et al., 2006*; *Samson et al., 2013*; *Zimmerman et al., 2020*). The effect of bacteriophages (viruses that infect bacteria) on their specific hosts and within the microbial community is largely determined by their lifestyle. Traditionally, most bacteriophage life cycles are classified as being either lytic or lysogenic (*Toit, 2017*). During the lytic cycle, phages hijack the host's metabolic machinery for replication of their own genome and the production of progeny particles that are released through lysis. In contrast, during lysogeny, phages integrate their genomes into the host

chromosome (or exist in the extrachromosomal form) and enter a dormant state, with the potential to re-enter a lytic cycle and release progeny by environmental changes at a later stage. Currently, more and more attention has been paid to chronic life cycles where bacterial growth continues despite phage reproduction (*Hoffmann Berling and Maze, 1964*), which was different from the lysogenic life cycle that could possibly lyse the host under some specific conditions. During a chronic cycle, progeny phage particles are released from host cells via extrusion or budding without killing the host (*Putzrath and Maniloff, 1977*; *Russel, 1991*; *Marvin et al., 2014*), instead of lysis of the host cell. Chronic infections are common among eukaryotic viruses (*Godkin and Smith, 2017*), however, to our best knowledge, only few phages have been described for prokaryotes in the pure isolates up to date (*Roux et al., 2019*; *Alarcón-Schumacher et al., 2022*; *Liu et al., 2022*). The best-studied examples of chronic infection are those from filamentous single-stranded DNA phages that to date have primarily been identified in the order *Tubulavirales*, which currently involves three families (*Inoviridae*, *Paulinoviridae*, and *Plectroviridae*) (*Ackermann, 2007*; *Ackermann, 2009*; *Knezevic et al., 2021*; *Chevallereau et al., 2022*). One of the most distinctive features of inoviruses is their ability to establish a chronic infection whereby their genomes may either integrate into the host chromosome or reside within the cell in a non-active form, and inoviruses particles are continuously released without killing the host. In addition, tailless phages enclosed in lipid membrane are also released from hosts during chronic life cycles (*Liu et al., 2022*). Recent studies indicate that chronic cycles are more widespread in nature than previously thought (*Mäntynen et al., 2021*), and the abundance of inoviruses in ecosystem may have been far underestimated (*Roux et al., 2019*). Therefore, a large percentage of phages in nature are proposed to replicate through chronic life cycles, but whether chronic cycles are associated with specific environments or ecological conditions remains to be thoroughly explored (*Chevallereau et al., 2022*).

Marine ecosystems always influence the operating conditions for life on earth via microbial interaction networks (*Falkowski et al., 1998*; *Faust and Raes, 2012*; *Wigington et al., 2016*), which are modulated by viruses through impacting their lifespan, gene flow, and metabolic outputs (*Suttle, 2005*). A majority of these viruses are bacteriophages, which exist widely in oceans and affect the life activities of microbes (*Breitbart, 2012*; *Roux et al., 2016*; *Gregory et al., 2019*; *Dominguez-Huerta et al., 2022*). In addition to core viral genes encoding viral structural proteins (*Brum et al., 2016*), bacteriophages also encode various auxiliary metabolic genes (AMGs) (*Brum and Sullivan, 2015*) that provide supplemental support to key steps of host metabolism. For instance, AMGs of marine bacteriophages have been predicted to be involved in photosynthesis (*Mann et al., 2003*), nitrogen cycling (*Ahlgren et al., 2019*; *Gazitúa et al., 2021*), sulfur cycling (*Anantharaman et al., 2014*; *Roux et al., 2016*), phosphorus cycling (*Zeng and Chisholm, 2012*), nucleotide metabolism (*Sullivan et al., 2005*; *Dwivedi et al., 2013*; *Enav et al., 2014*), and almost all central carbon metabolisms in host cells (*Hurwitz et al., 2013*). However, AMGs of chronic phages have not been reported. Thus, it is worth exploring whether chronic phages could carry AMGs and assist host metabolism during chronic infection.

The deep sea harbors abundant and undiscovered viruses, which potentially control the metabolism of microbial hosts and influence biogeochemical cycling (*Li et al., 2021*). However, due to the vast majority of deep-sea microbes cannot be cultivated in the laboratory, most bacteriophages could not be isolated. Thus, it is of great significance to further identify unknown phages in the deep sea and explore their relationship with microbial hosts and even marine ecosystems. Here, we report that polysaccharides can induce deep-sea *Lentisphaerae* bacteria (difficult-to-cultivate microorganisms) to release some chronic bacteriophages. These chronic bacteriophages might assist host polysaccharides metabolism via corresponding potential AMGs.

## Results and discussion
### Polysaccharides promote the growth of deep-sea *Lentisphaerae* strain WC36 and stimulate the expression of phage-associated genes

As a primary carbon source, polysaccharides are a ubiquitous energy source for microorganisms in both terrestrial and marine ecosystems (*Zheng et al., 2021a*). In our previous work, we successfully isolated a novel *Bacteroidetes* species through a polysaccharide degradation-driven strategy from the deep-sea cold seep (*Zheng et al., 2021a*). Of note, using the same approach and deep-sea sample

replicates, we also cultured a bacterial strain WC36 that was identified as a member of the phylum *Lentisphaerae* by 16S rRNA gene sequence-based phylogenetic analysis (*Figure 1A*). As expected, growth assays showed that two kinds of polysaccharides including laminarin and starch could promote strain WC36 growth (*Figure 1B*). In particular, supplementation of 10 g/l laminarin and 10 g/l starch in rich medium resulted in ~7- and ~4-fold growth promotion, respectively, compared to cultivation in rich medium (*Figure 1B*).

To explore the reasons behind this significant growth promotion by polysaccharide supplementation, we performed a transcriptomic analysis of strain WC36 cultured in rich medium supplemented either with or without 10 g/l laminarin for 5 and 10 days. In addition to the upregulation of genes related to glycan transport and degradation, when 10 g/l laminarin was added in the rich medium, the most upregulated genes were phage associated (e.g. phage integrase, phage portal protein) (*Figure 1C* and *Supplementary file 1*), which were expressed at the background level in the rich medium alone. Consistently, qRT-PCR results (*Figure 1D*) confirmed upregulation of some genes encoding phage-associated proteins, as shown in *Figure 1C*. Therefore, we speculate that the metabolism of polysaccharides in strain WC36 might be closely connected with the phage production.

## Polysaccharides induce the production of bacteriophages in *Lentisphaerae* strain WC36

To test our hypothesis that polysaccharides might induce strain WC36 to release bacteriophages, we grew strain WC36 in rich medium supplemented with or without laminarin or starch and isolated bacteriophages from the cell suspension supernatant according to established standard protocols (*Lin et al., 2012*). Based on the growth curve of strain WC36, we found that the growth rate of strictly anaerobic strain WC36 was relatively slow. Therefore, different to the typical sampling time (24 hr) for bacteriophage isolation from other bacteria (*Jiang and Paul, 1996*; *Weinbauer et al., 1999*), we selected a longer sampling time (10 days) to extract bacteriophages. Transmission electron microscopy (TEM) observation showed that many different shapes of phage-like structures indeed existed in the supernatant of WC36 cells cultured in rich medium supplemented with laminarin (*Figure 2A*, panels II–IV) or starch (*Figure 2A*, panels V–VIII). In contrast, we did not observe any phage-like structures in the supernatant of WC36 cells cultured in rich medium (*Figure 2A*, panel I). We also tested and confirmed that there were not any phage-like structures in rich medium supplemented with 10 g/l laminarin alone (*Figure 2—figure supplement 1A*) or in 10 g/l starch alone (*Figure 2—figure supplement 1B*), ruling out the possibility of phage contamination from the polysaccharides (laminarin/ starch). These results suggest that polysaccharides indeed stimulate the production of various bacteriophages from strain WC36. Correspondingly, in the presence of laminarin, different forms of bacteriophages were observed by TEM, with filamentous ones dominant (*Figure 2A*). The length of these filamentous phages was about ~0.4 to ~8.0 μm (*Figure 2A*, panel II), and the size of the hexagonal phages was less than 100 nm (*Figure 2A*, panel IV). In the presence of starch, in addition to filamentous and hexagonal phages (*Figure 2A*, panels V, VI, and VIII), we also observed a kind of icosahedral *Microviridae*-like phage with a diameter of 25–30 nm (*Figure 2A*, panel VII).

Given that we found bacteriophages in cultures of strain WC36, we next sought to explore whether bacteriophages adhering to bacterial cells could be observed. To this end, we checked the morphology of strain WC36 in the absence or presence of polysaccharides. As expected, with TEM we observed many filamentous phage-like structures around strain WC36 cells cultivated in rich medium supplemented with laminarin (*Figure 2B*, panel II) and starch (*Figure 2B*, panels III and IV). In contrast, we could not find any phage-like structures around strain WC36 cells cultivated in rich medium (*Figure 2B*, panel I). Meanwhile, we also checked the polysaccharides (laminarin/starch) in rich medium directly by TEM and did not find any phage-like structures (*Figure 2—figure supplement 2*). Moreover, based on TEM observation of ultrathin sections of strain WC36 cultured in medium supplemented with polysaccharide, we even observed filamentous phages that were being released from or entering into host cells via a kind of extrusion or budding structure around bacterial cells *Figure 2C*, panels II–IV; (*Figure 2—figure supplement 3B*, panels II and III), which were typical cellular structures in host cells used for chronic bacteriophage release (*Marvin et al., 2014*; *Krupovic, 2018*). In contrast, phage-like structures with extrusions or buddings were not found in cells of strain WC36 cultivated in rich medium (*Figure 2C*, panel I; *Figure 2—figure supplement 3B*, panel I). Hence, these results collectively suggest that filamentous bacteriophages induced from deep-sea strain WC36 replicate in

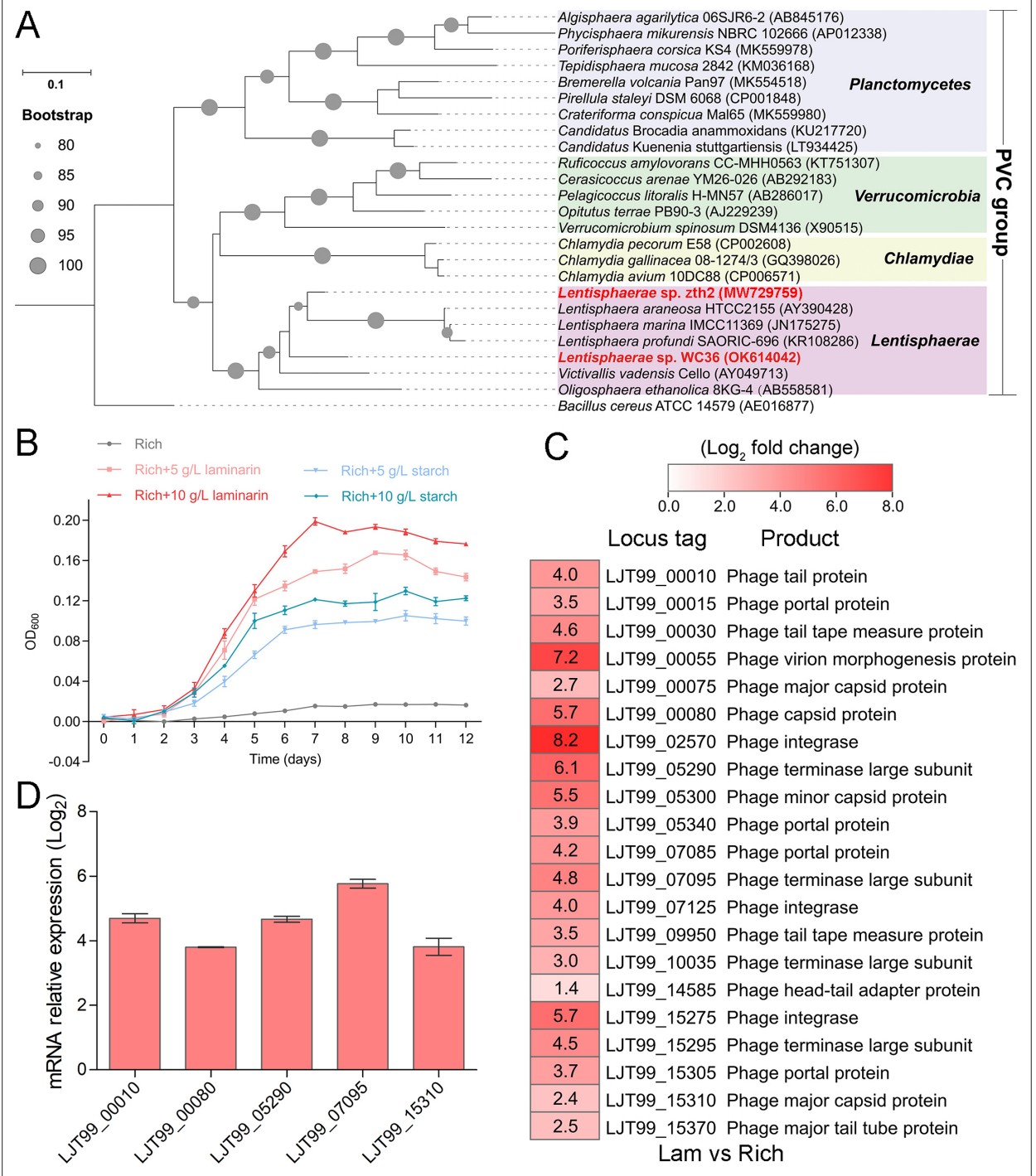

**Figure 1.** Polysaccharides promote the deep-sea *Lentisphaerae* strain WC36 growth and stimulate the expression of phage-associated genes. (**A**) Maximum likelihood phylogenetic tree of 16S rRNA gene sequences from strain WC36, strain zth2, and some *Planctomycetes–Verrucomicrobia–Chlamydia* (PVC) group bacteria. *Bacillus cereus* ATCC 14579 was used as the outgroup. Bootstrap values (%) >80 are indicated at the base of each node with the gray dots (expressed as percentages of 1000 replications). The accession number is displayed behind corresponding strain. (**B**) Growth assays of strain WC36 cultivated in rich medium either without supplementation, with 5 or 10 g/l laminarin or with 5 or 10 g/l starch. (**C**) Transcriptomics-based heat map showing all upregulated genes encoding phage-associated proteins in strain WC36 cultured in rich medium supplemented with 10 g/l laminarin. 'Rich' indicates strain WC36 cultivated in rich medium; 'Lam' indicates strain WC36 cultivated in rich medium supplemented with 10 g/l laminarin. (**D**) Quantitative real-time polymerase chain reaction (qRT-PCR) detection of the expression of some genes encoding phage-associated proteins shown in panel C. The heat map was generated by Heml 1.0.3.3. The numbers in panel C represent multiple differences in gene expression (by taking log₂ values).

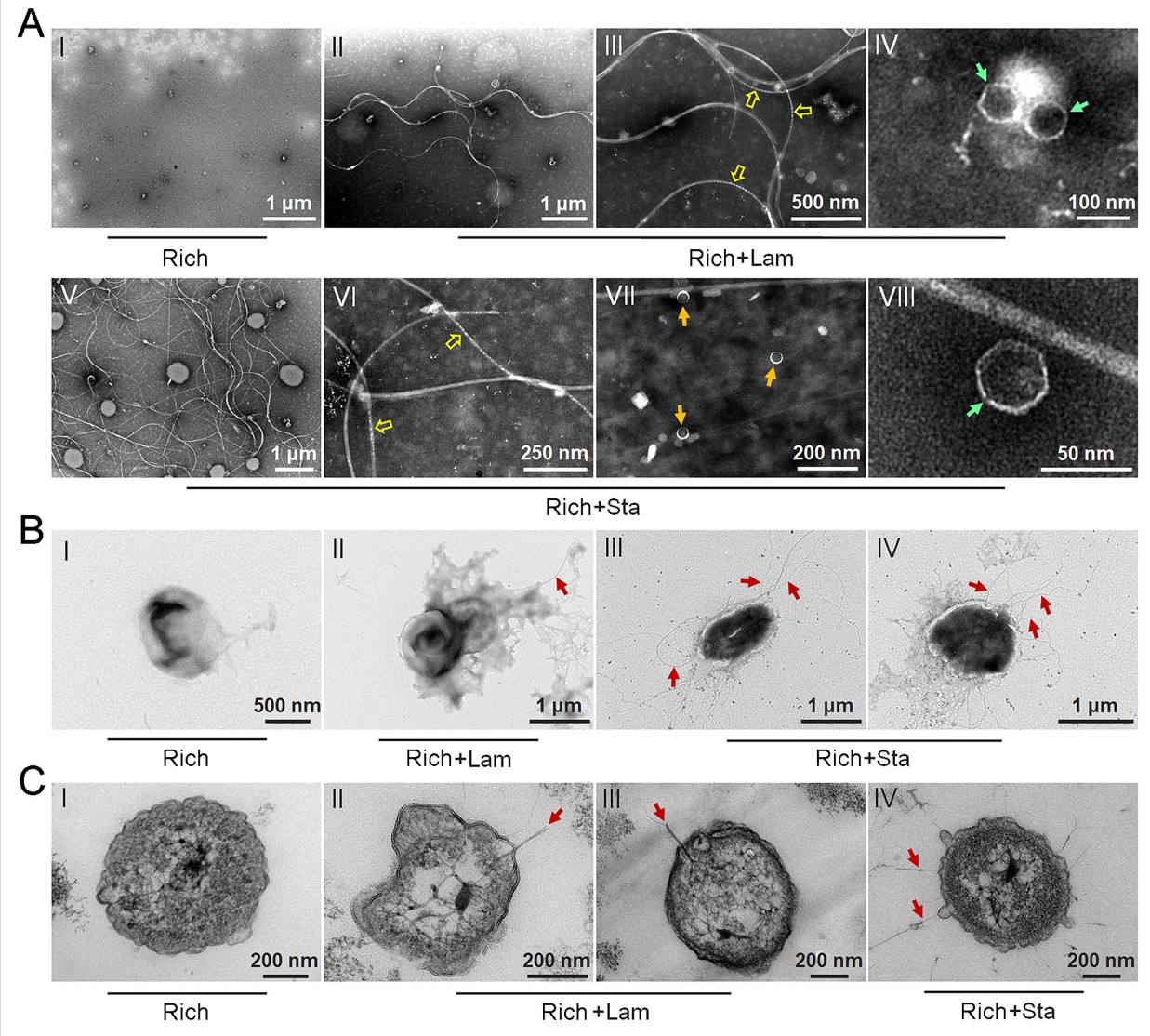

**Figure 2.** Polysaccharides induce the production of bacteriophages in *Lentisphaerae* strain WC36. (**A**) Transmission electron microscopy (TEM) observation of phages extracted from the supernatant of WC36 cells cultured in rich medium supplemented with or without polysaccharide. Panel I shows the absence of phages in the supernatant of WC36 cells cultivated in rich medium. Panels II–IV show the morphology of phages present in the supernatant of WC36 cells cultivated in rich medium supplemented with 10 g/l laminarin. Yellow hollow arrows indicate the typical granular structure of filamentous phages, and green arrows indicate phage-like particles with other shapes. Panels V–VIII show the morphology of phages present in the supernatant of strain WC36 cells cultivated in rich medium supplemented with 10 g/l starch. Typical filamentous phages are indicated with yellow hollow arrows, and two other kinds of phage-like particles with different shapes are indicated by orange and green arrows, respectively. (**B**) TEM observation of strain WC36 cultured in rich medium supplemented with or without 10 g/l laminarin or 10 g/l starch. Panel I shows representative morphology of strain WC36 cultivated in rich medium. Panel II shows the morphology of strain WC36 cultivated in rich medium supplemented with 10 g/l laminarin. Panels III and IV show the morphology of strain WC36 cultivated in rich medium supplemented with 10 g/l starch. Red arrows indicate filamentous phages associated with bacterial cells. (**C**) TEM of an ultrathin section of strain WC36 cultured in rich medium supplemented with or without 10 g/l laminarin or 10 g/l starch. Panel I shows an ultrathin section of strain WC36 cultivated in rich medium; panels II and III show ultrathin sections of strain WC36 cultivated in rich medium supplemented with 10 g/l laminarin; panel IV shows an observation of the ultrathin section of strain WC36 cultivated in rich medium supplemented with 10 g/l starch. Red arrows indicate filamentous phages being released from or entering bacterial cells.

The online version of this article includes the following figure supplement(s) for figure 2:

**Figure supplement 1.** Transmission electron microscopy observation of phage particles in the supernatant of different mediums.

**Figure supplement 2.** Transmission electron microscopy observation of different polysaccharides.

**Figure supplement 3.** The morphology of *Lentisphaerae* strain WC36 and its released filamentous bacteriophages.

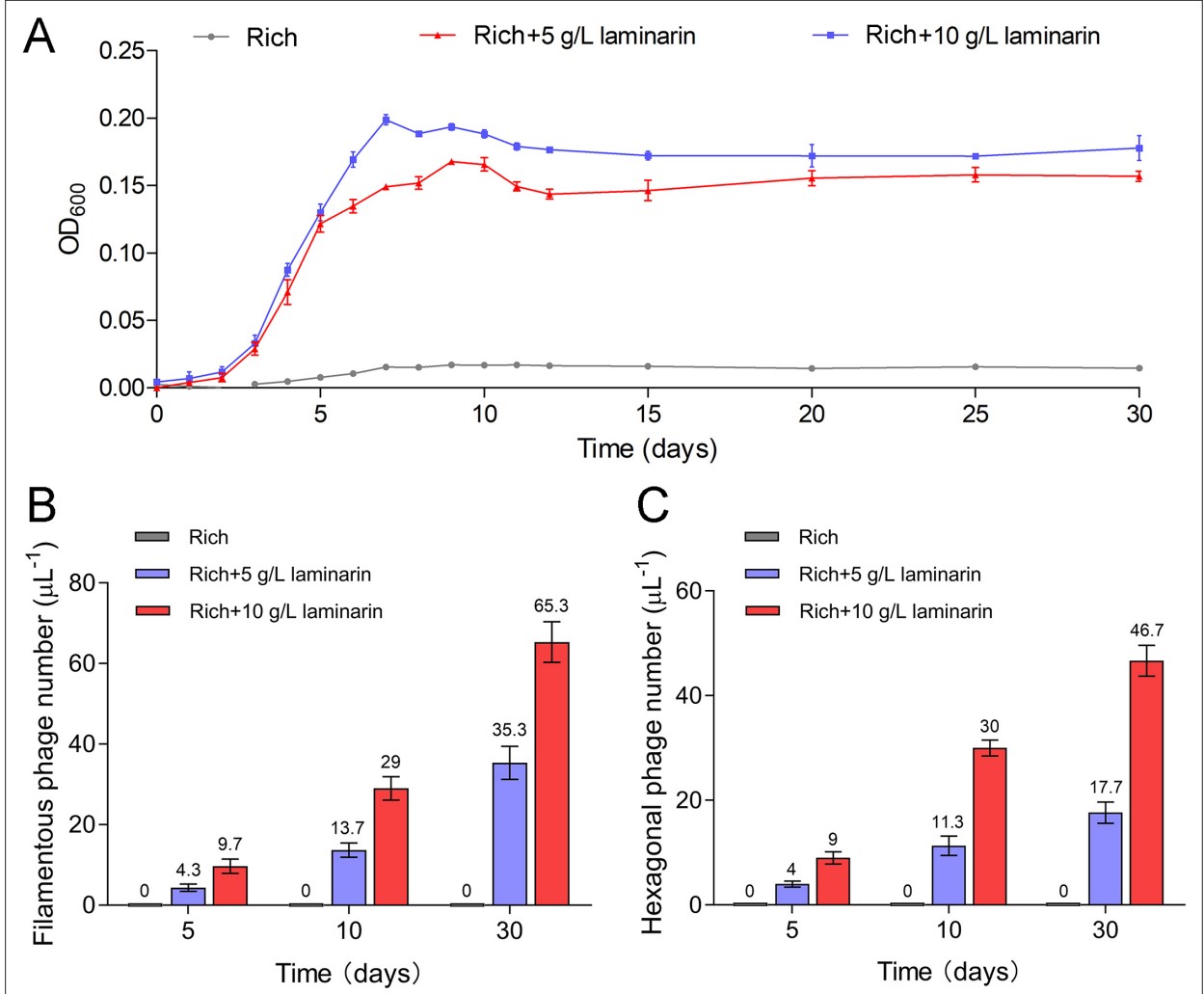

**Figure 3.** Bacteriophages induced from *Lentisphaerae* strain WC36 by polysaccharides are released via chronic manners. (**A**) Growth curve of strain WC36 cultivated in either rich medium alone or rich medium supplemented with 5 or 10 g/l laminarin for 30 days. The number of filamentous phages (**B**) and hexagonal phages (**C**) extracted from 1 μl of the supernatant of strain WC36 cell cultured in rich medium supplemented with or without 5 or 10 g/l laminarin (for 5, 10, and 30 days). The average numbers are shown at the top of the bar charts.

The online version of this article includes the following figure supplement(s) for figure 3:

**Figure supplement 1.** Transmission electron microscopy (TEM) observation of filamentous phages extracted from the supernatant of strain WC36 cell suspension cultured in rich medium supplemented with or without 5 or 10 g/l laminarin (for 5, 10, and 30 days).

a chronic manner as reported previously (*Chevallereau et al., 2022*; *Liu et al., 2022*). However, the entry and exit of the hexagonal phages into the WC36 cells were not observed.

To further verify the release style of bacteriophages from strain WC36 in the presence of polysaccharide, we prolonged the incubation time of this strain up to 30 days in medium supplemented with 5 or 10 g/l laminarin. Strain WC36 showed stable growth after it entered the stationary phase (*Figure 3A*). Regardless of whether the laminarin was present, the bacterial cells kept their cell shape intact, indicating they were still healthy after 30 days (*Figure 2—figure supplement 3A*, panels I–III). Apparently, the replication and release of bacteriophages in strain WC36 did not kill the host cell, consistent with the key feature of chronic bacteriophages (*Howard-Varona et al., 2017*). With an increase of culture time (5, 10, or 30 days) and laminarin concentration (5 or 10 g/l), the number of released bacteriophages increased (*Figure 2—figure supplement 3A*, panels IV–VI; *Figure 3— figure supplement 1*, panels IV–IX). Specifically, the average number per microliter of filamentous phages (9.7, 29, or 65.3) extracted from the supernatant of strain WC36 cultured in rich medium supplemented with 10 g/l laminarin for 5, 10, or 30 days was higher than that cultured in rich medium

supplemented with 5 g/l laminarin (4.3, 13.7, or 35.3) (*Figure 3B*). The average number per microliter of hexagonal phages (9, 30, and 46.7) extracted from the supernatant of strain WC36 cultured in rich medium supplemented with 10 g/l laminarin for 5, 10, or 30 days was higher than that cultured in rich medium supplemented with 5 g/l laminarin (4, 11.3, or 17.7) (*Figure 3C*).

## Polysaccharides promote the growth of deep-sea *Lentisphaerae* strain zth2 and induce bacteriophage production

To explore whether the production of bacteriophages induced by polysaccharide is an individual case, we further checked the effect of polysaccharides on another cultured deep-sea *Lentisphaerae* strain zth2. Strain zth2 was isolated by the same polysaccharide degradation-driven strategy as that used for strain WC36 (*Zhang et al., 2022*). 16S rRNA gene sequence similarity calculation using the NCBI server indicated that the closest relative of strain zth2 was strain WC36 (90.4%), which was lower than the threshold sequence identity (94.5%) for distinct genera (*Yarza et al., 2014*). Based on a maximum likelihood tree and similarity of 16S rRNA gene sequences, we propose that strain zth2 and WC36 are the type strains of two different novel genera, both belonging to the *Lentisphaeraceae* family, *Lentisphaerales* order, *Lentisphaeria* class. Consistently, laminarin indeed promoted the growth of strain zth2 (*Figure 4A*). Analysis of the differential transcriptome and qRT-PCR of strain zth2 cultured in either unsupplemented medium or with laminarin showed that many genes encoding critical factors associated with the phage life cycle were upregulated (ranging from 4- to 200-fold) (*Figure 4B, C* and *Supplementary file 2*), suggesting that phages might also be involved in the utilization of laminarin by strain zth2, the same as in strain WC36.

To further confirm the existence of bacteriophages in the supernatant of strain zth2 cell suspension cultured in medium supplemented with polysaccharide, we performed bacteriophage isolation from the supernatant of strain zth2 cultures followed by TEM. Indeed, many filamentous-, hexagonal-, and icosahedral phages were clearly observed in the culture of strain zth2 grown in the presence of laminarin (*Figure 4D*, panels I–III) and starch (*Figure 4D*, panels IV and V), but not in rich medium short of polysaccharide (*Figure 4D*, panel VI). This suggests that bacteriophages were induced by supplementation of polysaccharide to strain zth2.

In addition, transcriptome and qRT-PCR results showed that genes encoding bacterial secretion system-related proteins were upregulated in strains WC36 (*Figure 4—figure supplement 1A, B*) and zth2 (*Figure 4—figure supplement 1C, D*) cultured in medium supplemented with laminarin. It has been reported that filamentous phages utilize host cell secretion systems for their own egress from bacterial cells (*Davis et al., 2000*; *Bille et al., 2005*). Therefore, the filamentous phages induced from strains WC36 and zth2 might also get in and out of host cells using bacterial secretion systems. Taken together, we conclude that laminarin effectively promotes the growth of deep-sea *Lentisphaerae* strains WC36 and zth2, and induces the production of bacteriophages. Distinct from previous reports that bacteriophages are induced under traditional conditions such as low temperature (*Wang et al., 2007*), mitomycin C (*Mazaheri Nezhad Fard et al., 2010*), and ultraviolet irradiation (*McKay and Baldwin, 1973*), here, we report for the first time that polysaccharides induce the production of bacteriophages.

## Bacteriophages reprogram the polysaccharide metabolism of *Lentisphaerae* strains WC36 and zth2

In this study, we clearly show that polysaccharide effectively promotes bacterial growth and simultaneously induces the production of bacteriophages, suggesting a close relationship between bacteriophages and host polysaccharide metabolism. Therefore, we sought to ask whether bacteriophages could reprogram polysaccharide metabolism in deep-sea bacteria. For this purpose, we sequenced the genomes of the bacteriophages induced by polysaccharides in *Lentisphaerae* strains WC36 and zth2. Eventually, two incomplete assembled genomes (Phage-WC36-1, 6.3 kb; Phage-WC36-2, 28.3 kb) were obtained from the bacteriophages of strain WC36 (*Figure 5A, B* and *Supplementary file 3 and 4*). Meanwhile, two incomplete assembled genomes (Phage-zth2-1, 6.3 kb; Phage-zth2-2, 40.4 kb) were obtained from the phages of strain zth2 (*Figure 5A, C* and *Supplementary file 3 and 5*). Subsequently, we carefully compared the bacteriophage genomes with those of the corresponding hosts (strains WC36 and zth2) using Galaxy Version 2.6.0 (https://galaxy.pasteur.fr/) (*Afgan et al., 2018*) with the NCBI BLASTN method and used BWA-mem software for read mapping

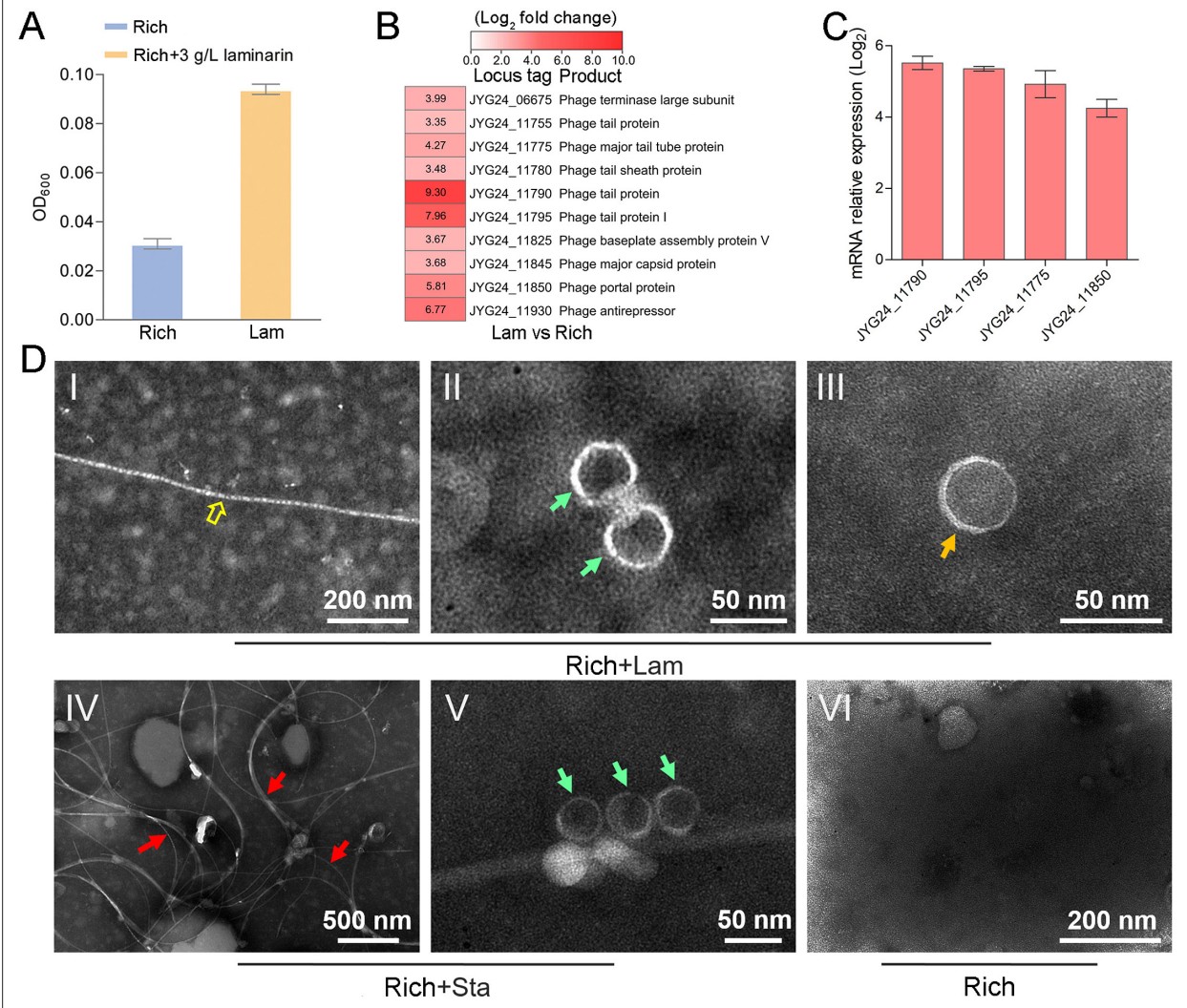

**Figure 4.** Polysaccharides promote the growth of deep-sea *Lentisphaerae* strain zth2 and induce the production of bacteriophages. (**A**) Growth assays of strain zth2 cultivated in rich medium supplemented with or without 3 g/l laminarin for 4 days. (**B**) Transcriptomics-based heat map showing all upregulated genes encoding phage-associated proteins in strain zth2 cultured in rich medium supplemented with 3 g/l laminarin. (**C**) qRT-PCR detection of the expressions of some genes encoding phage-associated proteins shown in panel B. The numbers in panel C represent multiple differences in gene expression (by taking log$_2$ values). (**D**) Transmission electron microscopy (TEM) observation of phages extracted from the supernatant of a cell suspension of strain zth2 cultured in the rich medium supplemented with or without polysaccharides. Panels I–III show the morphology of phages in the cell supernatant of strain zth2 cultivated in rich medium supplemented with 3 g/l laminarin. Typical filamentous phage is indicated with yellow hollow arrows, and two other kinds of phage-like particles with different shapes are indicated by orange and green arrows, respectively. Panels IV and V show the morphology of phages present in the cell supernatant of strain zth2 cultivated in rich medium supplemented with 3 g/l starch. Typical filamentous phages are indicated with red arrows, and other phage-like particles with different shapes are indicated by green arrows. Panel VI shows that no phages were observed in the cell supernatant of strain zth2 cultivated in rich medium. "Rich" indicates strain zth2 cultivated in rich medium; 'Lam' indicates strain zth2 cultivated in rich medium supplemented with 3 g/l laminarin.

The online version of this article includes the following figure supplement(s) for figure 4:

**Figure supplement 1.** Transcriptome profiles and qRT-PCR analysis of the genes encoding secretion system-related proteins.

from host whole-genome sequencing (WGS) to these bacteriophages. These analyses both showed that the bacteriophage genomes are completely outside of the host chromosomes. Through comparison of genome sequence (*Figure 5—figure supplement 1*), IMG database analysis, and phylogenetic analysis (*Figure 5—figure supplement 2*), we confirmed that Phage-WC36-1 was the same as Phage-zth2-1 (inoviruses, belonging to the family *Inoviridae*, order *Tubulavirales*, class *Faserviricetes*), indicating that inoviruses might be common in the *Lentisphaerae* phylum. Inovirus members were

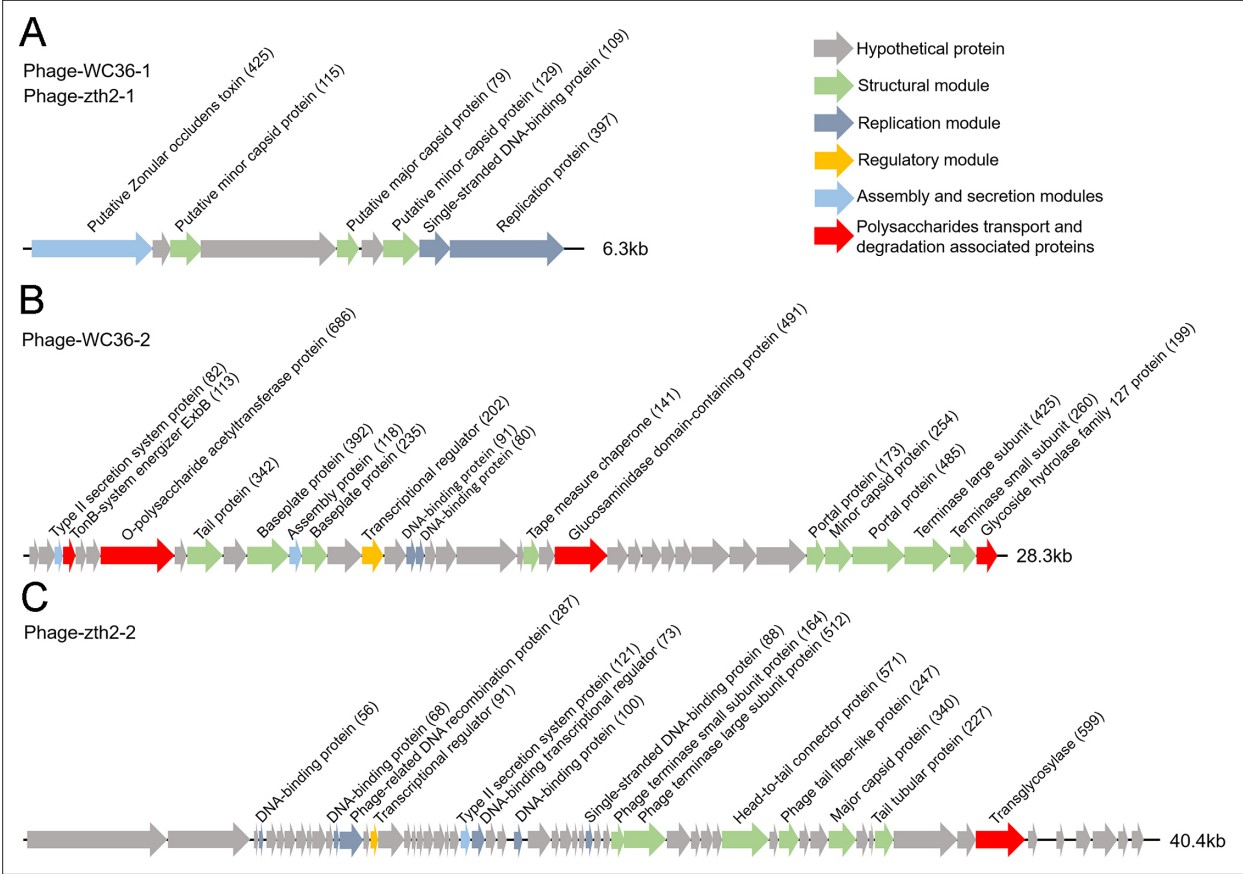

**Figure 5.** Genomic organization of bacteriophages released from *Lentisphaerae* strains WC36 and zth2 cultured in rich medium supplemented with polysaccharide. (**A**) Schematic of the genomic composition of Phage-WC36-1 and Phage-zth2-1. (**B**) A diagram showing the genomic composition of Phage-WC36-2. (**C**) The genomic composition of Phage-zth2-2. Arrows represent different ORFs (open reading frames) and the direction of transcription. The main putative gene products of the phages are shown, and the numbers in brackets indicate the numbers of amino acids within each ORF. Hypothetical proteins are indicated with gray arrows, structural modules are indicated by green arrows, the replication module is indicated by blue-gray arrows, the regulatory module is indicated by golden arrows, the assembly and secretion modules are indicated by blue arrows and potential auxiliary metabolic genes (AMGs) encoding proteins associated with polysaccharide transport and degradation are indicated by red arrows. The size of the phage genomes is shown behind each gene cluster.

The online version of this article includes the following figure supplement(s) for figure 5:

**Figure supplement 1.** Comparative genome analysis between the Phage-WC36-1 and Phage-zth2-1.

**Figure supplement 2.** Phylogenetic analysis of Phage-WC36-1 and related phages.

**Figure supplement 3.** Maximum likelihood phylogenetic tree of terL protein sequences from Phage-WC36-2, Phage-zth2-2, and some related *Caudoviricetes* phages.

the dominant population in the active group of viruses from deep-sea sediments, and encode some proteins contributing to host phenotypes and ultimately alter the fitness and other behaviors of their hosts (***Engelhardt et al., 2015***; ***Mai-Prochnow et al., 2015***). A maximum likelihood tree showed that both Phage-WC36-2 and Phage-zth2-2 belonged to the class *Caudoviricetes* (***Figure 5—figure supplement 3***). In addition to genes encoding phage-associated proteins within the assembled bacteriophage genomes, there are also many genes encoding proteins associated with polysaccharide transport and degradation, which were potential AMGs (***Shaffer et al., 2020***; ***Pratama et al., 2021***). These polysaccharide metabolizing proteins include TonB-system energizer ExbB (***Koebnik, 2005***), O-polysaccharide acetyltransferase (***Lunin et al., 2020***), glucosaminidase, glycoside hydrolase family 127 protein (best BLASTP hit *E*-value 3*e*−25), and transglycosylase (***Figure 5***). GH127 is one of the CAZymes, which are specifically responsible for degrading polysaccharides (***Cantarel et al., 2009***). Energizer ExbB is an important part of the TonB-system, which is often present in *Bacteroides* and responsible for transport of oligosaccharides from the outer membrane into the periplasm (***Foley***

*et al., 2016*). It is absent in the strain WC36 and zth2 genomes, however. In natural ecosystems, viral infections indirectly influence microbial metabolic fluxes, metabolic reprogramming, and energy homeostasis of host cells (*Howard-Varona et al., 2020*). For example, cyanoviruses possess AMGs encoding core photosynthetic reaction centers (*Sullivan et al., 2006*) which are highly expressed during infection, thus stimulating photosynthesis and enhancing viral abundance (*Lindell et al., 2007*). Therefore, these potential AMGs encoding TonB-system energizer ExbB and glycoside hydrolases might reprogram the polysaccharide metabolism of host cells (strains WC36 and zth2) by providing supplemental support to key metabolic processes. The above results provide stronger evidence that the relationship between Phages-WC36 and their host is not a fight to the death between enemies, but in contrast a mutualistic relationship between partners (*Shkoporov et al., 2022*). Overall, bacteriophages induced by polysaccharides contain potential AMGs associated with polysaccharide degradation, which may play a key role in the degradation and utilization of polysaccharide by host cells.

Moreover, it was recently demonstrated that selfish bacteria, which were common throughout the water column of the ocean, could bind, partially hydrolyze, and transport polysaccharides into the periplasmic space without loss of hydrolysis products (*Reintjes et al., 2017*; *Giljan et al., 2023*). Based on our results, we hypothesized that these chronic phages might assist the host to metabolize polysaccharides in this 'selfishness' mechanism, thus helping their hosts to grow and proliferate, so they could reap the benefits of simply having more hosts to infect. In addition, we found that nucleic acids could induce a deep-sea wall-less bacterium (*Hujiaoplasma nucleasis* zrk29) to release chronic bacteriophages, which might support strain zrk29 to metabolize nucleic acids without lysing host cells (*Zheng et al., 2023*). We also found that $NO_3^-$ or $NH_4^+$ induced a deep-sea *Planctomycetes* bacterium (*Poriferisphaera heterotrophicis* ZRK32) to release a bacteriophage in a chronic manner, without host cell lysis, and this chronic phage might enable strain ZRK32 to metabolize nitrogen through the function of AMGs (*Zheng et al., 2024*). Therefore, the mutualism between chronic phages and hosts may be widespread in the deep-sea microbes. However, it is difficult to conduct more experiments to prove the mechanistic roles of the AMGs in promoting both host and phage proliferation in these strictly anaerobic deep-sea microbes. In the future, we need to construct genetic operating systems of these strictly anaerobic host strains to reveal the relationship between chronic phage and host.

## Materials and methods
### Samples, media, and cultivation conditions

Deep-sea samples used for bacteria isolation in this study were collected from a typical cold seep (E119°17′07.322″, N22°06′58.598″) at a depth of 1146 m by *RV KEXUE* in July of 2018. Inorganic medium (including 1.0 g/l $NH_4Cl$, 1.0 g/l $NaHCO_3$, 1.0 g/l $CH_3COONa$, 0.5 g/l $KH_2PO_4$, 0.2 g/l $MgSO_4 \cdot 7H_2O$, 0.7 g/L cysteine hydrochloride, 500 µl/l 0.1% (wt/vol) resazurin, 1 l filtered seawater, pH 7.0) supplemented with 1.0 g/l of different polysaccharides (laminarin, fucoidan, starch, and alginate) under a 100% $N_2$ atmosphere was used to enrich microbes at 28°C for 1 month (*Zheng et al., 2021a*). The detailed isolation and purification steps were performed as described previously (*Zheng et al., 2021b*). After pure cultures of deep-sea bacteria were obtained, we cultured them in a rich medium (containing 1.0 g/l peptone, 5.0 g/l yeast extract, 1.0 g/l $NH_4Cl$, 1.0 g/l $NaHCO_3$, 1.0 g/l $CH_3COONa$, 0.5 g/l $KH_2PO_4$, 0.2 g/l $MgSO_4 \cdot 7H_2O$, 0.7 g/l cysteine hydrochloride, 500 µl/l 0.1% (wt/vol) resazurin, 1 l filtered seawater, pH 7.0) or rich medium supplemented with different polysaccharides for the future research.

### Genome sequencing, annotation, and analysis of strains WC36 and zth2

For genomic sequencing, strains WC36 and zth2 were grown in the liquid-rich medium supplemented with 5 g/l laminarin and starch and harvested after 1 week of incubation at 28°C. Genomic DNA was isolated by using the PowerSoil DNA isolation kit (Mo Bio Laboratories Inc, Carlsbad, CA). Thereafter, the genome sequencing was carried out with both the Illumina NovaSeq PE150 (San Diego, USA) and Nanopore PromethION platform (Oxford, UK) at the Beijing Novogene Bioinformatics Technology Co, Ltd. A complete description of the library construction, sequencing, and assembly was performed as previously described (*Zheng et al., 2021b*). We used seven databases to predict gene functions, including Pfam (Protein Families Database, http://pfam.xfam.org/), GO (Gene Ontology, http://gene-ontology.org/; *Ashburner et al., 2000*), KEGG (Kyoto Encyclopedia of Genes and Genomes, http://

www.genome.jp/kegg/; *Kanehisa et al., 2004*), COG (Clusters of Orthologous Groups, http://www.ncbi.nlm.nih.gov/COG/; *Galperin et al., 2015*), NR (Non-Redundant Protein Database databases), TCDB (Transporter Classification Database), and Swiss-Prot (http://www.ebi.ac.uk/uniprot/; *Bairoch and Apweiler, 2000*). A whole-genome Blast search (*E*-value less than 1$e$−5, minimal alignment length percentage larger than 40%) was performed against above seven databases.

## Phylogenetic analysis

The 16S rRNA gene tree was constructed based on full-length 16S rRNA gene sequences by the maximum likelihood method. The full-length 16S rRNA gene sequences of *Lentisphaerae* strains WC36 and zth2 were obtained from their genomes, and the 16S rRNA gene sequences of other related taxa used for phylogenetic analysis were obtained from NCBI (https://www.ncbi.nlm.nih.gov/). All the accession numbers are listed in *Supplementary file 6*. The maximum likelihood phylogenetic trees of Phage-WC36-1 were constructed based on the zona occludens toxin (Zot) and single-stranded DNA-binding protein (*Zeng et al., 2021*; *Evseev et al., 2023*). The maximum likelihood phylogenetic tree of Phage-WC36-2 and Phage-zth2-2 was constructed based on the terminase large subunit protein (*terL*). These proteins used to construct the phylogenetic trees were all obtained from the NCBI databases. All the sequences were aligned by MAFFT version 7 (*Katoh et al., 2019*) and manually corrected. The phylogenetic trees were constructed using the W-IQ-TREE web server (http://iqtree.cibiv.univie.ac.at) with the 'GTR+F+I+G4' model (*Trifinopoulos et al., 2016*). Finally, we used the online tool Interactive Tree of Life (iTOL v5) (*Letunic and Bork, 2021*) to edit the tree.

## Growth assay and transcriptomic analysis of *Lentisphaerae* strains WC36 and zth2 cultured in medium supplemented with laminarin

To assess the effect of polysaccharide on the growth of strain WC36, 100 µl of freshly incubated cells were inoculated in a 15-ml Hungate tube containing 10 ml rich medium supplemented with or without 5 or 10 g/l laminarin (or starch). To assess the effect of laminarin on the growth of strain zth2, 100 µl of freshly incubated cells were inoculated in a 15-ml Hungate tube containing 10 ml rich medium supplemented with or without 3 g/l laminarin. Each condition was performed in three replicates. All the Hungate tubes were incubated at 28°C. The bacterial growth status was monitored by measuring the $OD_{600}$ value via a microplate reader (Infinite M1000 Pro; Tecan, Mannedorf, Switzerland) every day until cell growth reached the stationary phase.

For transcriptomic analysis, a cell suspension of strain WC36 was harvested after culture in 1.5 l rich medium either without supplementation or with 10 g/l laminarin for 10 days. A cell suspension of strain zth2 was harvested after culture in rich medium either without supplementation or with 3 g/l laminarin for 4 days. Thereafter, these collected samples were used for transcriptomic analysis by Novogene (Tianjin, China). The detailed sequencing information was shown as below: (1) Library preparation for strand-specific transcriptome sequencing. A total amount of 3 µg RNA per sample was used as input material for the RNA sample preparations. Sequencing libraries were generated using NEBNext Ultra Directional RNA Library Prep Kit for Illumina (NEB, USA) according to the manufacturer's recommendations and index codes were added to attribute sequences to each sample. The rRNA is removed using a specialized kit that leaves the mRNA. Fragmentation was carried out using divalent cations under elevated temperature in NEBNext First Strand Synthesis Reaction Buffer (5×). First strand cDNA was synthesized using random hexamer primer and M-MuLV Reverse Transcriptase (RNaseH⁻). Second strand cDNA synthesis was subsequently performed using DNA Polymerase I and RNase H. In the reaction buffer, dNTPs with dTTP were replaced by dUTP. Remaining overhangs were converted into blunt ends via exonuclease/polymerase activities. After adenylation of 3′ ends of DNA fragments, NEBNext Adaptor with hairpin loop structure was ligated to prepare for hybridization. In order to select cDNA fragments of preferentially 150–200 bp in length, the library fragments were purified with AMPure XP system (Beckman Coulter, Beverly, USA). Then 3 µl USER Enzyme (NEB, USA) was used with size-selected, adaptor-ligated cDNA at 37°C for 15 min followed by 5 min at 95°C before PCR. Then PCR was performed with Phusion High-Fidelity DNA polymerase, Universal PCR primers and Index (X) Primer. At last, products were purified (AMPure XP system) and library quality was assessed on the Agilent Bioanalyzer 2100 system. (2) Clustering and sequencing. The clustering of the index-coded samples was performed on a cBot Cluster Generation System using TruSeq PE Cluster Kit v3-cBot-HS (Illumia) according to the manufacturer's instructions. After cluster generation,

the library preparations were sequenced on an Illumina HiSeq platform and paired-end reads were generated. (3) Data analysis. Raw data (raw reads) of fastq format were firstly processed through in-house perl scripts. In this step, clean data (clean reads) were obtained by removing reads containing adapter, reads containing ploy-N and low quality reads from raw data. At the same time, Q20, Q30, and GC content the clean data were calculated. All the downstream analyses were based on the clean data with high quality. Reference genome and gene model annotation files were downloaded from genome website directly. Both building index of reference genome and aligning clean reads to reference genome were used Bowtie2-2.2.3 (*Langmead and Salzberg, 2012*). HTSeq v0.6.1 was used to count the reads numbers mapped to each gene. And then FPKM of each gene was calculated based on the length of the gene and reads count mapped to this gene. FPKM, expected number of Fragments Per Kilobase of transcript sequence per Millions base pairs sequenced, considers the effect of sequencing depth and gene length for the reads count at the same time, and is currently the most commonly used method for estimating gene expression levels (*Anders and Huber, 2010*). (4) Differential expression analysis. Differential expression analysis of two conditions/groups (two biological replicates per condition) was performed using the DESeq R package (1.20.0) (*Wang et al., 2010*). DESeq provide statistical routines for determining differential expression in digital gene expression data using a model based on the negative binomial distribution. The resulting p-values were adjusted using the Benjamini and Hochberg's approach for controlling the false discovery rate. Genes with an adjusted $p < 0.05$ found by DESeq were assigned as differentially expressed. (For DEGSeq without biological replicates) Prior to differential gene expression analysis, for each sequenced library, the read counts were adjusted by edgeR program package through one scaling normalized factor. Corrected p-value of 0.005 and $\log_2$ (Fold change) of 1 were set as the threshold for significantly differential expression. (5) GO and KEGG enrichment analysis of differentially expressed genes. Gene Ontology (GO) enrichment analysis of differentially expressed genes was implemented by the GOseq R package, in which gene length bias was corrected (*Young et al., 2010*). GO terms with corrected p-value less than 0.05 were considered significantly enriched by differential expressed genes. KEGG is a database resource for understanding high-level functions and utilities of the biological system, such as the cell, the organism and the ecosystem, from molecular-level information, especially large-scale molecular datasets generated by genome sequencing and other high-throughput experimental technologies (http://www.genome.jp/kegg/; *Kanehisa et al., 2008*). We used KOBAS software to test the statistical enrichment of differential expression genes in KEGG pathways.

## Quantitative real-time polymerase chain reaction (qRT-PCR) assay

For qRT-PCR, cells of strain WC36 were cultured in basal medium supplemented with or without 10 g/l laminarin for 10 days, and cells of strain zth2 were cultured in basal medium supplemented without or with 3 g/l laminarin for 4 days. Total RNAs from each sample were extracted using the Trizol reagent (Solarbio, China) and the RNA concentration was measured using Qubit RNA Assay Kit in Qubit 2.0 Flurometer (Life Technologies, CA, USA). Then RNAs from corresponding sample were reverse transcribed into cDNA and the transcriptional levels of different genes were determined by qRT-PCR using SybrGreen Premix Low rox (MDbio, China) and the QuantStudioTM 6 Flex (Thermo Fisher Scientific, USA). The PCR condition was set as following: initial denaturation at 95°C for 3 min, followed by 40 cycles of denaturation at 95°C for 10 s, annealing at 60°C for 30 s, and extension at 72°C for 30 s. 16S rRNA was used as an internal reference and the gene expression was calculated using the $2^{-\Delta\Delta Ct}$ method, with each transcript signal normalized to that of 16S rRNA. Transcript signals for each treatment were compared to those of control group. Specific primers for genes encoding phage-associated proteins and 16S rRNA were designed using Primer 5.0 as shown in *Supplementary file 7*. All qRT-PCR runs were performed in three biological and three technical replicates.

## Bacteriophages isolation

Phage isolation was as described previously with some modifications (*Yamamoto et al., 1970*; *Tseng et al., 1990*; *Kim and Blaschek, 1991*). To induce the production of bacteriophages, strain WC36 was inoculated in rich medium supplemented with or without 10 g/l laminarin or 10 g/l starch, and strain zth2 was cultured in rich medium supplemented with or without 3 g/l laminarin or 3 g/l starch. After 10 days incubation, 60 ml of the different cultures was, respectively, collected and cells were removed by centrifugation at 8000 × *g*, 4°C for 20 min three times. The supernatant was filtered through a

0.22-µm millipore filter (Pall Supor, New York, America) and subsequently 1 M NaCl was added to lyse the bacteria and separate phage particles from these bacterial fragments. Then the supernatant was filtered through a 0.22-µm millipore filter and collected by centrifugation at 8000 × $g$, 4°C for 20 min, and phage particles were immediately precipitated with 100 g/l polyethylene glycol (PEG8000) at 4°C for 6 hr, and collected by centrifugation at 10,000 × $g$, 4°C for 20 min. The resulting phage particles were suspended in 2 ml suspension medium (SM) buffer (0.01% gelatin, 50 mM Tris–HCl, 100 mM NaCl, and 10 mM MgSO$_4$), then the suspension was extracted three times with an equal volume of chloroform (*Lin et al., 2012*) and collected by centrifugation at 4000 × $g$, 4°C for 20 min. Finally, clean phage particles were obtained.

## Transmission electron microscopy

To observe the morphology of bacteriophages, 10 µl phage virions were allowed to adsorb onto a copper grid for 20 min, and then stained with phosphotungstic acid for 30 s. Next, micrographs were taken with TEM (HT7700, Hitachi, Japan) with a JEOL JEM 12000 EX (equipped with a field emission gun) at 100 kV.

To observe the morphology of strain WC36, cell suspension in rich medium supplemented with or without polysaccharide was centrifuged at 5000 × $g$ for 10 min to obtain cell pellets. Subsequently, one part of the cell collection was adsorbed to the copper grids for 20 min, then washed with 10 mM phosphate buffer solution (PBS, pH 7.4) for 10 min and dried at room temperature for TEM observation. Another part was first preserved in 2.5% glutaraldehyde for 24 hr at 4°C, and then washed three times with PBS and dehydrated in ethanol solutions of 30%, 50%, 70%, 90%, and 100% for 10 min each time. Samples were further fixed in 1% osmium tetroxide for 2 hr and embedded in plastic resin (*Zechmann and Zellnig, 2009*; *Fortunato et al., 2016*). Thereafter, an ultramicrotome (Leica EM UC7, Germany) was used to prepare ultrathin sections (50 nm) of cells, and the obtained sections were stained with uranyl acetate and lead citrate (*Graham and Orenstein, 2007*; *Panphut et al., 2011*). Finally, all samples were observed with the JEOL JEM-1200 electron microscope as described above. In parallel, the phage' number in each condition was, respectively, calculated by ten random TEM images.

## Genome sequencing of bacteriophages

To sequence the genomes of bacteriophages, phage genomic DNA was extracted from different purified phage particles. Briefly, 1 µg/ml DNase I and RNase A were added to the concentrated phage solution for nucleic acid digestion overnight at 37°C. The digestion treatment was inactivated at 80°C for 15 min, followed by extraction with a Viral DNA Kit (Omega Bio-tek, USA) according to the manufacturer's instructions. Then, genome sequencing was performed by Biozeron Biological Technology Co Ltd (Shanghai, China). The detailed process of library construction, sequencing, genome assembly, and annotation was described below: (1) Library construction and Illumina HiSeq sequencing. Briefly, for Illumina pair-end sequencing of each phage, 1.0 µg genomic DNA was used for the sequencing library construction. Paired-end libraries with insert sizes of ~400 bp were prepared following the standard procedure. The purified genomic DNA was sheared into smaller fragments with a desired size by Covaris, and blunt ends were generated using the T4 DNA polymerase. And the desired fragments were purified through gel electrophoresis, then enriched and amplified by PCR. The index tag was introduced into the adapter at the PCR stage and we performed a library quality test. Finally, the qualified Illumina pair-end library was used for Illumina NovaSeq 6000 sequencing (150 bp*2, Shanghai BIOZERON Co, Ltd). (2) Genome assembly. The raw paired end reads were trimmed to remove the Illumina adaptors and to retain high-quality reads (score of >30 and length of >36 bases), as recommended by the Trimmomatic (version 0.36; *Bolger et al., 2014*) with parameters (SLIDINGWINDOW: 4:15, MINLEN: 75). Then clean data were obtained and used for further analysis. We have used the AByESS software (http://www.bcgsc.ca/platform/bioinfo/software/abyss) to perform genome assembly with multiple-Kmer parameters. VIBRANT v1.2.1 (*Kieft et al., 2020*), DRAM-v (*Shaffer et al., 2020*), VirSorter v1.0.5 (with categories 1 'pretty sure' and 2 'quite sure') (*Roux et al., 2015*) and VirFinder v1.1 (with statistically significant viral prediction: score >0.9 and p-value <0.05) (*Ren et al., 2017*) with default parameters were used to identify viral genomes from these assembly sequences by searching against the both cultured and non-cultured viral NCBI-RefSeq database (http://blast.ncbi.nlm.nih.gov/) and IMG/VR database (*Camargo et al., 2023*). The GapCloser software was subsequently

applied to fill up the remaining local inner gaps and correct the single base polymorphism for the final assembly results. The completeness of viral genomes was estimated using the CheckV v0.6.0 pipeline (*Nayfach et al., 2021*). (3) Genome annotation. For bacteriophages, these obtained genome sequences were subsequently annotated by searching these predicted genes against non-redundant (NR in NCBI, 20180814), SwissProt (release-2021_03, http://uniprot.org; *The UniProt Consortium, 2021*), KEGG (Release 94.0, http://www.genome.jp/kegg/) (*Kanehisa et al., 2021*), COG (update-2020_03, http://www.ncbi.nlm.nih.gov/COG, *Galperin et al., 2021*), and CAZy (update-2021_09, http://www.cazy.org/, *Drula et al., 2022*) databases. And the CAZymes were inferred from searches of the NCBI nonredundant (nr) protein database with BLASTP (https://blast.ncbi.nlm.nih.gov/), searches of UniProtKB with HMMer (hmmer.org) and searches of CAZy database with dbCAN tool (*Yin et al., 2012*; *Zhang et al., 2018*), using an *E*-value cutoff of $1 \times 10^{-6}$ for all three. Finally, all the results with the lowest *E*-value were marked as 'hypothetical protein'.

In order to detect whether these bacteriophage genomes were located within or outside of the host chromosomes, we used the new alignment algorithm BWA-MEM (Burrows-Wheeler aligner-maximum exact matches, version 0.7.15) (*Li, 2013*) with default parameters to perform read mapping of host WGS to these phages. In addition, we also evaluated the assembly graph underlying the host consensus assemblies. Clean reads were mapped to the bacterial complete genome sequences by Bowtie 2 (version 2.5.0) (*Langmead and Salzberg, 2012*), BWA (version 0.7.8) (*Li and Durbin, 2009*), and SAMTOOLS (version 0.1.18) (*Li et al., 2009*) with the default parameters.

## Acknowledgements

This work was funded by the NSFC Innovative Group Grant (No. 42221005), the Science and Technology Innovation Project of Laoshan Laboratory (Grant no. 2022QNLM030004-3 and LSKJ202203103), Major Research Plan of the National Natural Science Foundation (Grant no. 92351301), Shandong Provincial Natural Science Foundation (ZR2021ZD28 and ZR2023QD010), Strategic Priority Research Program of the Chinese Academy of Sciences (Grant no. XDA22050301), Key Collaborative Research Program of the Alliance of International Science Organizations (Grant no. ANSO-CR-KP-2022-08), and the Taishan Scholars Program (Grant nos. tstp20230637 and tsqn202312264).

## Additional information

### Funding

| Funder | Grant reference number | Author |
|---|---|---|
| NSFC Innovative Group | 42221005 | Chaomin Sun |
| Science and Technology Innovation Project of Laoshan Laboratory | 2022QNLM030004-3 | Chaomin Sun |
| Science and Technology Innovation Project of Laoshan Laboratory | LSKJ202203103 | Chaomin Sun |
| Major Research Plan of the National Natural Science Foundation | 92351301 | Chaomin Sun |
| Shandong Provincial Natural Science Foundation | ZR2021ZD28 | Chaomin Sun |
| Shandong Provincial Natural Science Foundation | ZR2023QD010 | Chong Wang |
| Strategic Priority Research Program of the Chinese Academy of Sciences | XDA22050301 | Chaomin Sun |

| Funder | Grant reference number | Author |
|---|---|---|
| Key Collaborative Research Program of the Alliance of International Science Organizations | ANSO-CR-KP-2022-08 | Chaomin Sun |
| Taishan Scholars Program | tstp20230637 | Chaomin Sun |
| Taishan Scholars Program | tsqn202312264 | Rikuan Zheng |

The funders had no role in study design, data collection, and interpretation, or the decision to submit the work for publication.

## Author contributions
Chong Wang, Conceptualization, Investigation, Methodology, Writing – original draft; Rikuan Zheng, Conceptualization, Investigation, Writing - review and editing; Tianhang Zhang, Resources, Writing - review and editing; Chaomin Sun, Conceptualization, Supervision, Funding acquisition, Writing - review and editing

## Author ORCIDs
Rikuan Zheng (iD) https://orcid.org/0009-0007-0275-0592
Chaomin Sun (iD) https://orcid.org/0000-0003-4802-184X

Reviewer #1 (Public Review): https://doi.org/10.7554/eLife.92345.3.sa1
Reviewer #2 (Public Review): https://doi.org/10.7554/eLife.92345.3.sa2
Author response https://doi.org/10.7554/eLife.92345.3.sa3

---

# Additional files

## Supplementary files
• Supplementary file 1. Annotations and fold-changes of the transcriptome data of *Lentisphaerae* strain WC36 cultured in rich medium supplemented with or without 10 g/l laminarin. 'Rich' indicates strain WC36 cultivated in rich medium; 'Lam' indicates strain WC36 cultivated in rich medium supplemented with 10 g/l laminarin.

• Supplementary file 2. Annotations of the transcriptome data of Lentisphaerae strain zth2 cultured in rich medium supplemented with or without 3 g/l laminarin. 'Rich' indicates strain zth2 cultivated in rich medium; 'Lam' indicates strain zth2 cultivated in rich medium supplemented with 3 g/l laminarin.

• Supplementary file 3. Annotations of the Phage-WC36-1 and Phage-zth2-1 genome.

• Supplementary file 4. Annotations of the Phage-WC36-2 genome.

• Supplementary file 5. Annotations of the Phage-zth2-2 genome.

• Supplementary file 6. Accession numbers of strain WC36, strain zth2, some *Planctomycetes–Verrucomicrobia–Chlamydia* (PVC) group bacteria, and *Bacillus cereus* ATCC 14579.

• Supplementary file 7. Primers used for qRT-PCR.

• MDAR checklist

## Data availability
The complete genome sequences of strains WC36 and zth2 presented in this study have been deposited in the GenBank database with accession numbers CP085689 and CP071032, respectively. The whole 16S rRNA gene sequences of strains WC36 and zth2 have been deposited in the GenBank database with accession numbers OK614042 and MW729759, respectively. The genome sequences of Phage-WC36-1 (Phage-zth2-1), Phage-WC36-2, and Phage-zth2-2 have been deposited in the GenBank database with accession numbers PP701473, OL791266, and OL791268, respectively. The raw sequencing reads from the transcriptomics analysis have been deposited to the NCBI Short Read Archive (accession numbers: PRJNA946146 and PRJNA756144).

The following datasets were generated:

| Author(s) | Year | Dataset title | Dataset URL | Database and Identifier |
|---|---|---|---|---|
| Wang C, Zheng RK, Sun CM | 2021 | Lentisphaerae bacterium WC36 chromosome, complete genome | https://www.ncbi.nlm.nih.gov/nuccore/CP085689 | NCBI GenBank, CP085689 |
| Sun CM, Zhang TH | 2021 | Lentisphaerota bacterium strain zth2 chromosome, complete genome | https://www.ncbi.nlm.nih.gov/nuccore/CP071032 | NCBI GenBank, CP071032 |
| Wang C, Zheng RK, Sun CM | 2021 | Lentisphaerae bacterium strain WC36 16S ribosomal RNA gene, partial sequence | https://www.ncbi.nlm.nih.gov/nuccore/OK614042 | NCBI Nucleotide, OK614042 |
| Zhang T, Sun C | 2021 | Victivallis sp. strain zth2 16S ribosomal RNA gene, partial sequence | https://www.ncbi.nlm.nih.gov/nuccore/MW729759 | NCBI Nucleotide, MW729759 |
| Wang C | 2024 | UNVERIFIED_ORG: Phage WC36-1-filamentous, partial genome | https://www.ncbi.nlm.nih.gov/nuccore/PP701473 | NCBI Nucleotide, PP701473 |
| Wang C | 2022 | UNVERIFIED: Phage WC36-2 genomic sequence | https://www.ncbi.nlm.nih.gov/nuccore/OL791266 | NCBI Nucleotide, OL791266 |
| Wang C, Zheng KR, Sun MC | 2022 | UNVERIFIED: Phage zth2-2, complete genome | https://www.ncbi.nlm.nih.gov/nuccore/OL791268 | NCBI Nucleotide, OL791268 |
| Zheng Ri | 2023 | Transcriptome analysis of strain WC36 | https://www.ncbi.nlm.nih.gov/bioproject/PRJNA946146/ | NCBI BioProject, PRJNA946146 |
| Zhang T | 2021 | Transcriptome sequencing of Glycandegcoldseepsis marina zth2 | https://www.ncbi.nlm.nih.gov/bioproject/PRJNA756144/ | NCBI BioProject, PRJNA756144 |

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
