## [Editor Report · eLife assessment]

This manuscript presents **valuable** findings on two isolates of deep sea Lentisphaerae strains, which further our understanding of deep sea microbial life. The manuscript's primary claim is that phage isolates augment polysaccharide use in Pseudomonas bacteria, with preliminary evidence for the potential auxiliary metabolic genes in chronic phage infection and/or host proliferation. The strength of the evidence is overall **solid** and there are only minor weaknesses regarding the mechanism of polysaccharide use by the phages and the evidence for chronic infection. Overall, the data on Lentisphaerae strains will deepen our understanding of microbial life in the deep sea.

---

## [Referee Report · Reviewer #1 (Public Review)]

Summary:

I have previously reviewed this manuscript as a submission to another journal in 2022. My recommendations here mirror those of my prior suggestions, now with further added details.

This manuscript describes the identification and isolation of several phage from deep sea isolates of Lentisphaerae strains WC36 and zth2. The authors observe induction of several putative chronic phages with the introduction of additional polysaccharides to the media. The authors suggest that two of the recovered phage genomes encode AMGs associated with polysaccharide use. The authors also suggest that adding the purified phage to cultures of Pseudomonas stutzeri 273 increased the growth of this bacteria due to augmented polysaccharide use genes from the phage.

Strengths:

Interesting isolate of deep sea Lentisphaerae strains which will undoubtedly further our understanding of deep sea microbial life.

The revisions have addressed the weaknesses raised in the previous review.

---

## [Referee Report · Reviewer #2 (Public Review)]

Summary:

This paper investigates deep-sea bacteriophage systems which appear to employ a chronic replication mechanism that is induced or enhanced by polysaccharide addition. Some preliminary evidence for the potential role of auxiliary metabolic genes in aiding phage and/or host proliferation is also provided. The hypothesis being tested is fully supported with solid and convincing evidence and the findings are potentially generalizable with implications for our understanding of polysaccharide-mediated virus-host interactions and carbon cycling in marine ecosystems more broadly.

Strengths:

This paper synthesizes sequencing and phylogenic analyses of two Lentisphaerae bacteria and three phage genomes; electron microscopy imaging of bacterial/phage particles; differential gene expression analyses; differential growth curve analyses, and differential phage proliferation assays to extract insights into whether laminarin and starch can induce both host growth and phage proliferation. The data presented convincingly demonstrate that both host culture density and phage proliferation increase as a result having host, phage, and polysaccharide carbon source together in culture.

Weaknesses:

The AMG-centered elements of the article would be strengthened by more "mechanistic" experiments focusing on identifying "HOW" the polysaccharide processing, transport, and metabolism genes are being used by the phages to either directly increase viral infection/replication or else to indirectly do so by supporting the growth of the host (via mutualism). The concept of "selfishness" in bacterial systems and its potential role in viral life cycles could be more developed. Selfish bacteria are active throughout the water column of the ocean. ISME COMMUN. 3, 11 (2023) (see for instance https://doi.org/10.1038/s43705-023-00219-7) and such "selfish" bacteria sequester metabolizable polysaccharides in their periplasm to advantage (https://www.nature.com/articles/ismej201726). It is plausible that phages may be either hijacking such polysaccharide sequestration mechanisms to improve infectivity and ENTRY or else helping their hosts to grow and proliferate so they can reap the benefits of simply having more hosts to infect. The current work does not clearly distinguish between these two distinct mechanistic possibilities. The paper would be strengthened by a more detailed/clear discussion of this possibility.

---

## [Author Response]

The following is the authors’ response to the original reviews.

**eLife assessment**
This manuscript presents useful findings on several phage from deep sea isolates of *Lentisphaerae* strains WC36 and zth2 that further our understanding of deep sea microbial life. The manuscript's primary claim is that phage isolates augment polysaccharide use in *Pseudomonas* bacteria via auxiliary metabolic genes (AMGs). However, the strength of the evidence is incomplete and does not support the primary claims. Namely, there are not data presented to rule out phage contamination in the polysaccharide stock solution, AMGs are potentially misidentified, and there is missing evidence of successful infection.

Thanks for the Editor’s and Reviewers’ positive and constructive comments, which help us improve the quality of our manuscript entitled “Deep-sea bacteriophages facilitate host utilization of polysaccharides” (paper#eLife-RP-RA-2023-92345). The comments are valuable, and we have studied the comments carefully and have made corresponding revisions according to the suggestions. We removed some uncertain results and strengthened other parts of the manuscript, which evidently improved the accuracy and impact of the revised version. Revised portions are marked in blue in the modified manuscript. Please find the detailed responses as following.

**Public Reviews:**

**Reviewer #1 (Public Review):**
Summary: This manuscript describes the identification and isolation of several phage from deep sea isolates of *Lentisphaerae* strains WC36 and zth2. The authors observe induction of several putative chronic phages with the introduction of additional polysaccharides to the media. The authors suggest that two of the recovered phage genomes encode AMGs associated with polysaccharide use. The authors also suggest that adding the purified phage to cultures of *Pseudomonas stutzeri* 273 increased the growth of this bacterium due to augmented polysaccharide use genes from the phage. While the findings were of interest and relevance to the field, it is my opinion that several of the analysis fall short of supporting the key assertions presented.

Thanks for your comments. We removed some uncertain results and strengthened other parts of the manuscript, which evidently improved the accuracy and impact of the revised version. Please find the detailed responses as following.

Strengths: Interesting isolate of deep sea *Lentisphaerae* strains which will undoubtedly further our understanding of deep sea microbial life.

Thanks for your positive comments.

Weaknesses:(1) Many of the findings are consistent with a phage contamination in the polysaccharide stock solution.

Thanks for your comments. We are very sure that the phages are specifically derived from the *Lentisphaerae* strain WC36 but not the polysaccharide stock solution. The reasons are as following: (1) the polysaccharide stock solution was strictly sterilized to remove any phage contamination; (2) we have performed multiple TEM checks of the rich medium supplemented with 10 g/L laminarin alone (Supplementary Fig. 1A) or in 10 g/L starch alone (Supplementary Fig. 1B), and there were not any phage-like structures, which confirmed that the polysaccharides (laminarin/starch) we used were not contaminated with any phage-like structures; in addition, we also observed the polysaccharides (laminarin/starch) directly by TEM and did not find any phage-like structures (Supplementary Fig. 2); (3) the polysaccharide (starch) alone could not promote the growth of *Pseudomonas stutzeri* 273, however, the supplement of starch together with the extracted Phages-WC36 could effectively facilitate the growth of *Pseudomonas stutzeri* 273 (Response Figure 1). The above results clearly indicated the phages were derived from the *Lentisphaerae* strain WC36 but not the polysaccharide stock solution.

**Author response image 1. sa3fig1:** Growth curve and status of *Pseudomonas stutzeri* 273 cultivated in basal medium, basal medium supplemented with 20 μl/mL Phages-WC36, basal medium supplemented with 5 g/L starch, basal medium supplemented with 5 g/L starch and 20 μl/mL Phages-WC36.

(2) The genes presented as AMGs are largely well known and studied phage genes which play a role in infection cycles.

Thanks for your comments. Indeed, these AMGs may be only common in virulent phages, while have never been reported in chronic phages. In virulent phages, these genes typically act as lysozymes, facilitating the release of virions from the host cell upon lysis, or injection of viral DNA upon infection. However, the chronic phages do not lyse the host. Therefore, the persistence of these genes in chronic phages may be due to their ability to assist the host in metabolizing polysaccharides. Finally, according to your suggestions, we have weakened the role of AMGs and added “potential” in front of it. The detailed information is shown below.

(3) The evidence that the isolated phage can infect *Pseudomonas stutzeri* 273 is lacking, putting into question the dependent results.

Thanks for your comments. Actually, we selected many marine strains (*Pseudomonadota*,
*Planctomycetes*, *Verrucomicrobia*, *Fusobacteria*, and *Tenericutes* isolates) to investigate whether Phages-WC36 could assist them in degradation and utilization of polysaccharides, and found that Phages-WC36 could only promote the growth of strain 273. It is reported that filamentous phages could recognize and bind to the host pili, which causes the pili to shrink and brings the filamentous phages closer to and possibly through the outer membrane of host cells. The possible mechanism of other chronic phages release without breaking the host might be that it was enclosed in lipid membrane and released from the host cells by a nonlytic manner. Thus, these chronic phages may have a wider host range. However, we were unable to further reveal the infection mechanism due to some techniques absence. Therefore, according to your suggestions, we have deleted this section in the revised manuscript.

**Reviewer #1 (Recommendations For The Authors):**
I have previously reviewed this manuscript as a submission to another journal in 2022. My recommendations here mirror those of my prior suggestions, now with further added details.

Thanks for your great efforts for reviewing our manuscript and valuable suggestions for last and this versions.

Specific comments:Comment 1: Line 32. Rephrase to "polysaccharides cause the induction of multiple temperate phages infecting two strains of *Lentisphaerae* (WC36 and zth2) from the deep sea."

Thanks for your positive suggestion. We have modified this description as “Here, we found for the first time that polysaccharides induced the production of multiple temperate phages infecting two deep-sea *Lentisphaerae* strains (WC36 and zth2).” in the revised manuscript (Lines 31-33).

Comment 2: Line 66. "Chronic" infections are not "lysogenic" as described here, suggesting the former is a subcategory of the latter. If you are going to introduce lifecycles you need a brief sentence distinguishing "chronic" from "lysogenic"

Thanks for your positive suggestion. We added this sentence as “Currently, more and more attention has been paid to chronic life cycles where bacterial growth continues despite phage reproduction (Hoffmann Berling and Maze, 1964), which was different from the lysogenic life cycle that could possibly lyse the host under some specific conditions.” in the revised manuscript (Lines 66-69).

Comment 3: Line 72. Please avoid generalized statements like "a hand-full" (or "plenty" line 85). Try to be at least somewhat quantitative regarding how many chronic phages are known. This is a fairly common strategy among archaeal viruses.

Thanks for your suggestion. Given that some filamentous phages also have a chronic life cycle that is not explicitly reported, we cannot accurately estimate their numbers. According to your suggestions, we have modified these descriptions as “however, to our best knowledge, only few phages have been described for prokaryotes in the pure isolates up to date (Roux et al., 2019; Alarcón-Schumacher et al., 2022; Liu et al., 2022).” in the revised manuscript (Lines 73-75). In addition, the number of chronic phages in the biosphere cannot be accurately estimated, according to the latest report (Chevallereau et al., 2022), which showed that “a large fraction of phages in the biosphere are produced through chronic life cycles”. Therefore, we have modified this description as “Therefore, a large percentage of phages in nature are proposed to replicate through chronic life cycles” in the revised manuscript (Lines 87-88).

Comment 4: Line 93. While Breitbart 2012 is a good paper to cite here, there have been several, much more advanced analysis of the oceans virome. https://doi.org/10.1016/j.cell.2019.03.040 is one example, but there are several others. A deeper literature review is required in this section.

Thanks for your valuable suggestions. We have added some literatures and modified this description as “A majority of these viruses are bacteriophages, which exist widely in oceans and affect the life activities of microbes (Breitbart, 2012; Roux et al., 2016; Gregory et al., 2019; Dominguez-Huerta et al., 2022).” in the revised manuscript (Lines 94-97).

References related to this response:

Roux, S., Brum, J.R., Dutilh, B.E., Sunagawa, S., Duhaime, M.B., Loy, A., Poulos, B.T., Solonenko, N., Lara, E., Poulain, J., et al. (2016) Ecogenomics and potential biogeochemical impacts of globally abundant ocean viruses. *Nature* 537:689-693.

Gregory, A.C., Zayed, A.A., Conceição-Neto, N., Temperton, B., Bolduc, B., Alberti, A., Ardyna, M., Arkhipova, K., Carmichael, M., Cruaud, C., et al. (2019) Marine DNA Viral Macro- and Microdiversity from Pole to Pole. *Cell* 177:1109-1123.e1114.

Dominguez-Huerta, G., Zayed, A.A., Wainaina, J.M., Guo, J., Tian, F., Pratama, A.A., Bolduc, B., Mohssen, M., Zablocki, O., Pelletier, E., et al. (2022) Diversity and ecological footprint of Global Ocean RNA viruses. *Science* 376:1202-1208.

Comment 5: Line 137. I see the phage upregulation in Figure 1, however in the text and figure it would be good to also elaborate on what the background expression generally looks like. Perhaps a transcriptomic read normalization and recruitment to the genome with a display of the coverage map, highlighting the prophage would be helpful. Are the polysacharides directly influencing phage induction or is there some potential for another cascading effect?

Thanks for your comments. We have elaborated all expressions of phage-associated genes under different conditions in the Supplementary Table 1, which showed that the background expressions were very low. The numbers in Fig. 1C were the gene expressions (by taking log2 values) of strain WC36 cultured in rich medium supplemented with 10 g/L laminarin compared with the rich medium alone.

In addition, our qRT-PCR results (Fig. 1D) also confirmed that these genes encoding phage-associated proteins were significantly upregulated when 10 g/L laminarin was added in the rich medium. According to your suggestions, we have modified this description as “In addition to the up-regulation of genes related to glycan transport and degradation, when 10 g/L laminarin was added in the rich medium, the most upregulated genes were phage-associated (e. g. phage integrase, phage portal protein) (Fig. 1C and Supplementary Table 1), which were expressed at the background level in the rich medium alone.” in the revised manuscript (Lines 136-140). Based on the present results, we speculate that polysaccharides might directly induce phage production, which needs to be verified by a large number of experiments in the future.

Comment 6: Line 179. We need some assurance that phage was not introduced by your laminarin or starch supplement. Perhaps a check on the TEM/sequencing check of supplement itself would be helpful? This may be what is meant on Line 188 "without culturing bacterial cells" however this is not clearly worded if that is the case. Additional note, further reading reinforces this as a key concern. Many of the subsequent results are consistent with a contaminated starch stock.

Thanks for your comments. We are very sure that the phages are specifically derived from the *Lentisphaerae* strain WC36 but not the polysaccharide stock solution. The reasons are as following: (1) we have performed multiple TEM checks of the rich medium supplemented with 10 g/L laminarin alone (Supplementary Fig. 1A) or in 10 g/L starch alone (Supplementary Fig. 1B), and there were not any phage-like structures, which confirmed that the polysaccharides (laminarin/starch) we used are not contaminated with any phage-like structures. In addition, we also observed the polysaccharides (laminarin/starch) directly by TEM and did not find any phage-like structures (Supplementary Fig. 2). According to your suggestions, we have modified this description as “We also tested and confirmed that there were not any phage-like structures in rich medium supplemented with 10 g/L laminarin alone (Supplementary Fig. 1A) or in 10 g/L starch alone (Supplementary Fig. 1B), ruling out the possibility of phage contamination from the polysaccharides (laminarin/ starch).” in the revised manuscript (Lines 158-162) and “Meanwhile, we also checked the polysaccharides (laminarin/ starch) in rich medium directly by TEM and did not find any phage-like structures (Supplementary Fig. 2).” in the revised manuscript (Lines 178-180). (2) the polysaccharide stock solution was strictly sterilized to remove any phage contamination. (3) the polysaccharide (starch) alone could not promote the growth of *Pseudomonas stutzeri* 273, however, the supplement of starch together with the extracted Phages-WC36 could effectively facilitate the growth of *Pseudomonas stutzeri* 273 (Response Figure 1). The above results clearly indicated the phage was derived from the *Lentisphaerae* strain WC36 but not the polysaccharide stock solution.

In addition, given that polysaccharide was a kind of critical energy source for most microorganisms, we sought to ask whether polysaccharide also induces the production of bacteriophages in other deep-sea bacteria. To this end, we cultured deep-sea representatives from other four other phyla (including *Chloroflexi*, *Tenericutes*, *Proteobacteria*, and *Actinobacteria*) in the medium supplemented with laminarin/starch, and checked the supernatant of cells suspension through TEM as described above. We could not find any phage-like structures in these cells suspension (Response Figure 2), which also confirmed that there was no phage contamination in the polysaccharides.

**Author response image 2. sa3fig2:** Growth curve and status of *Pseudomonas stutzeri* 273 cultivated in basal medium, basal medium supplemented with 20 μl/mL Phages-WC36, basal medium supplemented with 5 g/L starch, basal medium supplemented with 5 g/L starch and 20 μl/mL Phages-WC36.

**Author response image 3. sa3fig3:** TEM observation of the supernatant of cells suspension of a *Chloroflexi* strain, a *Tenericutes* strain, a *Proteobacteria* strain and an *Actinobacteria* strain that cultivated in the rich medium supplemented with 10 g/L laminarin and 10 g/L starch. No phage-like particles could be observed.

Comment 7: Line 223. Correct generalized wording "long time".

Thanks for your comments. We have changed “after for a long time” to “after 30 days” in the revised manuscript (Line 197).

Comment 8: Line 229. Please more explicitly describe what these numbers are (counts of virion like structures - filamentous and hexagonal respectively?), the units (per µL?), and how these were derived. The word "around" should be replaced with mean and standard deviation values for each count from replicates, without which these are not meaningful.

Thanks for your comments. The average numbers per microliter (µL) of filamentous and hexagonal phages in each condition were respectively calculated by randomly choosing ten TEM images. According to your suggestions, we have modified this description as “Specifically, the average number per microliter of filamentous phages (9.7, 29 or 65.3) extracted from the supernatant of strain WC36 cultured in rich medium supplemented with 10 g/L laminarin for 5, 10 or 30 days was higher than that cultured in rich medium supplemented with 5 g/L laminarin (4.3, 13.7 or 35.3) (Fig. 3B). The average number per microliter of hexagonal phages (9, 30, 46.7) extracted from the supernatant of strain WC36 cultured in rich medium supplemented with 10 g/L laminarin for 5, 10 or 30 days was higher than that cultured in rich medium supplemented with 5 g/L laminarin (4, 11.3 or 17.7) (Fig. 3C).” in the revised manuscript (Lines 203-210).

Comment 9: Line 242. This section should be included in the discussion of Figure 2 - around line 194.

Thanks. According to your suggestion, we have moved this section to the discussion corresponding to Figure 2 (Lines 183-191).

Comment 10: Figure 3. Stay consistent in the types of figures generated per strain. Figure 3A should be a growth curve.

Thanks for your comments. Actually, figure 3A was a growth curve, the corresponding description “(A) Growth curve of strain WC36 cultivated in either rich medium alone or rich medium supplemented with 5 g/L or 10 g/L laminarin for 30 days.” was shown in the Figure 3A legend in this manuscript.

Comment 11: Line 312. Move the discussion of AMGs to after the discussion of the phage genome identification.

Thanks for your valuable comments. According to your suggestions, we have moved the discussion of AMGs to after the discussion of the phage genome identification.

Comment 12: Line 312. It would be informative to sequence in-bulk each of your treatments as opposed to just sequencing the viral isolates (starch and no host included) to see what viruses can be identified in each. ABySS is also not a common assembler for viral analysis. Is there literature to support it as a sufficient tool in assembling viral genomes? What sequencing depths were obtained in your samples?

Thanks for your comments. In previous studies, we did sequence the starch or laminarin alone (no host included) and did not detect any phage-related sequences. The introduction of ABySS software was shown in these literatures (Jackman SD, Vandervalk BP, Mohamadi H, Chu J, Yeo S, Hammond SA, Jahesh G, Khan H, Coombe L, Warren RL, Birol I. ABySS 2.0: resource-efficient assembly of large genomes using a Bloom filter. Genome Res. 2017 May;27(5):768-777; Simpson JT, Wong K, Jackman SD, Schein JE, Jones SJ, Birol I. ABySS: a parallel assembler for short read sequence data. Genome Res. 2009 Jun;19(6):1117-23.), which were also used to assemble viral genomes in these literatures (Guo Y, Jiang T. First Report of *Sugarcane Mosaic Virus* Infecting Goose Grass in Shandong Province, China. Plant Dis. 2024 Mar 21. doi: 10.1094/PDIS-11-23-2514-PDN; Tang M, Chen Z, Grover CE, Wang Y, Li S, Liu G, Ma Z, Wendel JF, Hua J. Rapid evolutionary divergence of Gossypium barbadense and G. hirsutum mitochondrial genomes. BMC Genomics. 2015 Oct 12;16:770.). The sequencing depth of the phages of strain WC36 and zth2 were 350x and 365x, respectively.

Comment 13: Line 323. Replace "eventually" with more detail about what was done to derive the genomes. Were these the only four sequences identified as viral?

Thanks for your comments. We have used the ABySS software (http://www.bcgsc.ca/platform/bioinfo/software/abyss) to perform genome assembly with multiple-Kmer parameters. VIBRANT v1.2.1 (Kieft et al., 2020), DRAM-v (Shaffer et al., 2020), VirSorter v1.0.5 (with categories 1 (“pretty sure”) and 2 (“quite sure”)) (Roux et al., 2015) and VirFinder v1.1 (with statistically significant viral prediction: score > 0.9 and *P*-value < 0.05) (Ren et al., 2017) with default parameters were used to identify viral genomes from these assembly sequences by searching against the both cultured and non-cultured viral NCBI-RefSeq database (http://blast.ncbi.nlm.nih.gov/) and IMG/VR database (Camargo et al., 2023). The GapCloser software was subsequently applied to fill up the remaining local inner gaps and correct the single base polymorphism for the final assembly results. All the detailed processes were described in the supplementary information. The virus sequences with higher scores are only these four, but they are not complete genomes. Some virus sequences with shorter sequences and lower scores were excluded.

Comment 14: Line 328. We need some details about the host genomes here. How were these derived? What is their completeness/contamination? What is their size? If the bins are poor, these would not serve as a reliable comparison to identify integrated phage.

Thanks for your comments. For genomic sequencing, strains WC36 and zth2 were grown in the liquid rich medium supplemented with 5 g/L laminarin and starch and harvested after one week of incubation at 28 °C. Genomic DNA was isolated by using the PowerSoil DNA isolation kit (Mo Bio Laboratories Inc, Carlsbad, CA). Thereafter, the genome sequencing was carried out with both the Illumina NovaSeq PE150 (San Diego, USA) and Nanopore PromethION platform (Oxford, UK) at the Beijing Novogene Bioinformatics Technology Co., Ltd. A complete description of the library construction, sequencing, and assembly was performed as previously described (Zheng et al., 2021). We used seven databases to predict gene functions, including Pfam (Protein Families Database, http://pfam.xfam.org/), GO (Gene Ontology, http://geneontology.org/) (Ashburner et al., 2000), KEGG (Kyoto Encyclopedia of Genes and Genomes, http://www.genome.jp/kegg/) (Kanehisa et al., 2004), COG (Clusters of Orthologous Groups, http://www.ncbi.nlm.nih.gov/COG/) (Galperin et al., 2015), NR (Non-Redundant Protein Database databases), TCDB (Transporter Classification Database), and Swiss-Prot (http://www.ebi.ac.uk/uniprot/) (Bairoch and Apweiler, 2000). A whole genome Blast search (E-value less than 1e-5, minimal alignment length percentage larger than 40%) was performed against above seven databases.

The completeness of the genomes of strains WC36 and zth2 were 100%, which were checked by the CheckM v1.2.2. The size of the genome of strains WC36 and zth2 were 3,660,783 bp and 3,198,720bp, respectively. The complete genome sequences of strains WC36 and zth2 presented in this study have been deposited in the GenBank database with accession numbers CP085689 and CP071032, respectively.

Moreover, to verify whether the absence of microbial contamination in phage sequencing results, we used the new alignment algorithm BWA-MEM (version 0.7.15) to perform reads mapping of host WGS to these phages. We found that all the raw reads of host strains (WC36 and zth2) were not mapping to these phages sequences (Response Figure 3, shown as below). In addition, we also performed the evaluation of the assembly graph underlying the host consensus assemblies. Clean reads were mapped to the bacterial complete genome sequences by the Bowtie 2 (version 2.5.0), BWA (version 0.7.8) and SAMTOOLS (version 0.1.18). The results showed that the total mismatch rate of strains WC36 and zth2 were almost 0% and 0.03%, respectively (Response Table 1, shown as below). In addition, we also collected the cells of strains WC36 and zth2, and then sent them to another company for whole genome sequencing (named WC36G and ZTH, GenBank accession numbers CP151801 and CP119760, respectively). The completeness of the genomes of strains

WC36G and ZTH were also 100%. The size of the genome of strains WC36G and

ZTH were 3,660,783bp and 3,198,714bp, respectively. The raw reads of strains WC36G and zth2 were also not mapping to the phages sequences. Therefore, we can confirm that these bacteriophage genomes were completely outside of the host chromosomes.

**Author response image 4. sa3fig4:** The read mapping from WGS to phage sequences.

**Author response table 1. sa3table1:** Sequencing depth and coverage statistics.

ref_name	query_name	avg_depth	coverage >= *1X*	coverage >= *4X*	coverage >= *10X*	coverage >= *20X*
WC36	WC36.illumina	229	100	100	100	100
zth2	zth2.illumina	274	100	100	100	100
WC36	WC36.illumina	100	0	848834700	848834149	551
zth2	zth2.illumina	99.97	0.03	877980000	877716606	263394

References related to this response:

Zheng, R., Liu, R., Shan, Y., Cai, R., Liu, G., and Sun, C. (2021b) Characterization of the first cultured free-living representative of *Candidatus* Izemoplasma uncovers its unique biology *ISME J* 15:2676-2691.

Ashburner, M., Ball, C.A., Blake, J.A., Botstein, D., Butler, H., Cherry, J.M., Davis, A.P., Dolinski, K., Dwight, S.S., Eppig, J.T., et al. (2000) Gene ontology: tool for the unification of biology. The Gene Ontology Consortium *Nat Genet* 25:25-29.

Kanehisa, M., Goto, S., Kawashima, S., Okuno, Y., and Hattori, M. (2004) The KEGG resource for deciphering the genome *Nucleic Acids Res* 32:D277-280.

Galperin, M.Y., Makarova, K.S., Wolf, Y.I., and Koonin, E.V. (2015) Expanded microbial genome coverage and improved protein family annotation in the COG database *Nucleic Acids Res* 43:D261-269.

Bairoch, A., and Apweiler, R. (2000) The SWISS-PROT protein sequence database and its supplement TrEMBL in 2000 *Nucleic Acids Res* 28:45-48.

Comment 15: Line 333. This also needs some details. What evidence do you have that these are not chromosomal? If not chromosomal where can they be found? Sequencing efforts should also be able to yield extrachromosomal elements such as plasmids etc... If you were to sequence your purified isolate cultures from the rich media alone and include all assemblies (not just those binned for example) as a reference, would you be able to recruit viral reads? The way this reads suggests that Chevallereau et al., worked specifically with these phage, which is not the case - please rephrase.

Thanks for your comments. We carefully compared the bacteriophage genomes with those of the corresponding hosts (strains WC36 and zth2) using Galaxy Version 2.6.0 (https://galaxy.pasteur.fr/) (Afgan et al., 2018) with the NCBI BLASTN method and used BWA-mem software for read mapping from host whole genome sequencing (WGS) to these bacteriophages. These analyses both showed that the bacteriophage genomes are completely outside of the host chromosomes. Therefore, we hypothesized that the phage genomes might exist in the host in the form similar to that of plasmid.

Comment 16: Line 335. More to the point here that we need confirmation that these phages were not introduced in the polysaccharide treatment

Thanks for your comments. Please find our answers for this concern in the responses for comment 1 of “weakness” part and comment 6 of “Recommendations For The Authors” part.

Comment 17: Line 342. Lacking significant detail here. Phylogeny based on what gene(s), how were the alignments computed/refined, what model used etc..?

Thanks for your comments. According to your suggestions, all the related information was shown in this section “Materials and methods” of this manuscript. The maximum likelihood phylogenetic tree of Phage-WC36-2 and Phage-zth2-2 was constructed based on the terminase large subunit protein (terL). These proteins used to construct the phylogenetic trees were all obtained from the NCBI databases. All the sequences were aligned by MAFFT version 7 (Katoh et al., 2019) and manually corrected. The phylogenetic trees were constructed using the W-IQ-TREE web server (http://iqtree.cibiv.univie.ac.at) with the “GTR+F+I+G4” model (Trifinopoulos et al., 2016). Finally, we used the online tool Interactive Tree of Life (iTOL v5) (Letunic and Bork, 2021) to edit the tree.

Comment 18: Line 346. How are you specifically defining AMGs in this study? Most of these are well-known and studied phage genes with specific life cycle functions and could not be considered as polysaccharide processing AMGs even though in host cells many do play a role in polysaccharide processing systems. A substantially deeper literature review is needed in this section, which would ultimately eliminate most of these from the potential AMG pools. Further, the simple HMM/BLASTp evalues are not sufficient to support the functional annotation of these genes. At a minimum, catalytic/conserved regions should be identified, secondary structures compared, and phylogenetic analysis (where possible) developed etc... My recommendation is to eliminate this section entirely from the manuscript.Categorically:- Glycoside hydrolase (various families), glucosaminidases, and transglycosylase are all very common to phage and operate generally as a lysins, facilitating the release of virions from the host cell upon lysis, or injection of viral DNA upon infection https://doi.org/10.3389/fmicb.2016.00745 (and citations therein) https://doi.org/10.1016/j.cmi.2023.10.018 etc... In order to confirm these as distinct AMGs we would need a very detailed analysis indicating that these are not phage infection cycle/host recognition related, however I strongly suspect that under such interrogation, these would prove to be as such.-TonB related systems including ExbB are well studied among phages as part of the trans-location step in infection. These could not be considered as AMGs. https://doi.org/10.1128/JB.00428-19. Other TonB dependent receptors play a role in host recognition.-Several phage acetyltransferases play a role in suppressing host RNA polymerase in order to reserve host cell resources for virion production, including polysaccharide production. https://doi.org/10.3390/v12090976. Further it has been shown that the *E. coli* gene *neuO* (O-acetyltransferase) is a homologue of lambdoid phage tail fiber genes https://doi.org/10.1073/pnas.0407428102. I suspect the latter is also the case here and this is a tail fiber gene.

Thanks for your valuable comments. According to your suggestions, we have reanalyzed these AMGs and made some modifications (the new version Fig. 5A, shown as below). These genes encoding proteins associated with polysaccharide transport and degradation may be only common in virulent phages, and have never been reported in chronic phages. Unlike virulent phages, these genes typically act as lysozymes, facilitating the release of virions from the host cell upon lysis, or injection of viral DNA upon infection, chronic phages do not lyse the host. It is reported that, filamentous phages could recognize and bind to the host pili, which causes the pili to shrink and brings the filamentous phages closer to and possibly through the outer membrane of host cells (Riechmann *et al*., 1997; Sun *et al*., 1987). The possible mechanism of other chronic phage release without breaking the host might be that it was enclosed in lipid membrane and released from the host cells by a nonlytic manner. It has recently been reported that the tailless *Caudoviricetes* phage particles are enclosed in lipid membrane and are released from the host cells by a nonlytic manner (Liu et al., 2022), and the prophage induction contributes to the production of membrane vesicles by *Lacticaseibacillus casei* BL23 during cell growth (da Silva Barreira et al., 2022). Therefore, the persistence of these genes in chronic phages may be due to their ability to assist the host in metabolizing polysaccharides.

Finally, according to your suggestions, we have weakened the role of AMGs and added “potential” in front of it.

References related to this response:

Riechmann L, Holliger P. (1997) The C-terminal domain of TolA is the coreceptor for filamentous phage infection of *E. coli Cell* 90:351-60.

Sun TP, Webster RE. (1987) Nucleotide sequence of a gene cluster involved in entry of E colicins and single-stranded DNA of infecting filamentous bacteriophages into *Escherichia coli J Bacteriol* 169:2667-74.

Liu Y, Alexeeva S, Bachmann H, Guerra Martníez J.A, Yeremenko N, Abee T et al. (2022) Chronic release of tailless phage particles from *Lactococcus lactis Appl Environ Microbiol* 88: e0148321. da Silva Barreira, D., Lapaquette, P., Novion Ducassou, J., Couté, Y., Guzzo, J., and Rieu, A. Spontaneous prophage induction contributes to the production of membrane vesicles by the gram-positive bacterium *Lacticaseibacillus casei* BL23. mBio_._ 2022;13:e0237522.

Comment 19: Line 354. To make this statement that these genes are missing from the host, we would need to know that these genomes are complete.

Thanks for your comments. The completeness of the genomes of strains WC36 and zth2 were 100%, which were checked by the CheckM v1.2.2. The size of the genome of strains WC36 and zth2 were 3,660,783 bp and 3,198,720bp, respectively. The complete genome sequences of strains WC36 and zth2 presented in this study have been deposited in the GenBank database with accession numbers CP085689 and CP071032, respectively. In addition, we also collected the cells of strains WC36 and zth2, and then sent it to another company for whole genome sequencing (named WC36G and ZTH, GenBank accession numbers CP151801 and CP119760, respectively). The completeness of the genomes of strains WC36G and ZTH were also 100%. The size of the genome of strains WC36G and ZTH were 3,660,783bp and 3,198,714bp, respectively. Therefore, these genomes of strains WC36 and zth2 were complete and circular.

Comment 20: Figure 5. Please see https://peerj.com/articles/11447/ and https://doi.org/10.1093/nar/gkaa621 for a detailed discussion on vetting AMGs. Several of these should be eliminated according to the standards set in the field. More specifically, and by anecdotal comparison with other inoviridae genomes, for Phage-WC36-1 and Phage-zth2-1, I am not convinced that the transactional regulator and glycoside hydrolase are a part of the phage genome. The phage genome probably ends at the strand switch.

Thanks for your comments. According to your suggestions, we have analyzed these two articles carefully and modified the genome of Phage-WC36-1 and Phage-zth2-1 by anecdotal comparison with other inoviridae genomes. As you said, the transactional regulator and glycoside hydrolase are not a part of the phage genome.

The new version Fig. 5A was shown.

References related to this response:

Shaffer, M., Borton, M.A., McGivern, B.B., Zayed, A.A., La Rosa, S.L., Solden, L.M., Liu, P., Narrowe, A.B., Rodrgíuez-Ramos, J., Bolduc, B., et al. (2020) DRAM for distilling microbial metabolism to automate the curation of microbiome function *Nucleic Acids Res* 48:8883-8900

Pratama, A.A., Bolduc, B., Zayed, A.A., Zhong, Z.P., Guo, J., Vik, D.R., Gazitúa, M.C., Wainaina, J.M., Roux, S., and Sullivan, M.B. (2021) Expanding standards in viromics: in silico evaluation of dsDNA viral genome identification, classification, and auxiliary metabolic gene curation *PeerJ* 9:e11447

Comment 21: Line 380. This section needs to start with detailed evidence that this phage can even infect this particular strain. Added note, upon further reading the serial dilution cultures are not sufficient to prove these phage infect this Pseudomonas. We need at a minimum a one-step growth curve and wet mount microscopy. It is much more likely that some carry over contaminant is invading the culture and influencing OD600. With the given evidence, I am not at all convinced that these phages have anything to do with *Pseudomonas* polysaccharide use and I recommend either drastically revising this section or eliminating it entirely.Line 386-389. Could this be because you are observing your added phage in the starch enriched media while no phage were introduced with the "other types of media" so none would be observed? This could have nothing to do with infection dynamics. Further, this would also be consistent with your starch solution being contaminated by phage.Line 399. Again consistent with the starch media being contaminated.Line 401-408. This is more likely to do with the augmentation of the media with an additional carbon source and not involving the phage.Line 410. I am not convinced that these viruses infect the *Pseudomonas* strain. Extensive further evidence of infection is needed to make these assertions. Figure 6A. We need confirmation that the isolate culture remains pure and there are no other contaminants introduced with the phage.

Thanks for your comments. We have proved that the polysaccharides (laminarin/ starch) didn't contaminate any phages above. Actually, we selected many marine strains (*Pseudomonadota*,
*Planctomycetes*, *Verrucomicrobia*, *Fusobacteria*, and *Tenericutes* isolates) to investigate whether Phages-WC36 could assist them in degradation and utilization of polysaccharides, and found that Phages-WC36 could only promote the growth of strain 273. The presence of filamentous phages and hexagonal phages was detected in the supernatant of strain 273 cultured in basal medium supplemented with 5 g/L starch and 20 μl/mL Phages-WC36. After 3 passages of serial cultivation in basal medium supplemented with 5 g/L starch, we found that filamentous phages and hexagonal phages were also present in basal medium supplemented with starch, but not in the basal medium, which may mean that Phages-WC36 could infect strain 273 and starch is an important inducer. In addition, the Phages-WC36 used in the growth assay of strain 273 were multiple purified and eventually suspended in SM buffer (0.01% gelatin, 50 mM Tris-HCl, 100 mM NaCl and 10 mM MgSO4). Thus, these phages are provided do not contain some extracellular enzymes and/or nutrients. In addition, we set up three control groups in the growth assay of strain 273: basal medium, basal medium supplemented with Phages-WC36 and basal medium supplemented with starch. If the Phages-WC36 contains some extracellular enzymes and/or nutrients, strain 273 could also grow well in the basal medium supplemented only with Phages-WC36. However, the poor growth results of strain 273 cultivated in the basal medium supplemented with Phages-WC36 further confirmed that there were not some extracellular enzymes and/or nutrients in these phages.

Finally, the possible mechanism of the chronic phage release without breaking the host might be that it was enclosed in lipid membrane and released from the host cells by a nonlytic manner. Thus, these chronic phages may have a wider host range. However, we were unable to further disclose the infection mechanism in this paper. Therefore, according to your suggestions, we have deleted this section entirely.

Comment 27: Line 460. Details about how these genomes were reconstructed is needed here.

Thanks for your comments. According to your suggestions, we have added the detailed information about the genome sequencing, annotation, and analysis as “Genome sequencing, annotation, and analysis of strains WC36 and zth2 For genomic sequencing, strains WC36 and zth2 were grown in the liquid rich medium supplemented with 5 g/L laminarin and starch and harvested after one week of incubation at 28 °C. Genomic DNA was isolated by using the PowerSoil DNA isolation kit (Mo Bio Laboratories Inc, Carlsbad, CA). Thereafter, the genome sequencing was carried out with both the Illumina NovaSeq PE150 (San Diego, USA) and Nanopore PromethION platform (Oxford, UK) at the Beijing Novogene Bioinformatics Technology Co., Ltd. A complete description of the library construction, sequencing, and assembly was performed as previously described (Zheng et al., 2021b). We used seven databases to predict gene functions, including Pfam (Protein Families Database, http://pfam.xfam.org/), GO (Gene Ontology, http://geneontology.org/) (Ashburner et al., 2000), KEGG (Kyoto Encyclopedia of Genes and Genomes, http://www.genome.jp/kegg/) (Kanehisa et al., 2004), COG

(Clusters of Orthologous Groups, http://www.ncbi.nlm.nih.gov/COG/) (Galperin et al.,2015), NR (Non-Redundant Protein Database databases), TCDB (Transporter Classification Database), and Swiss-Prot (http://www.ebi.ac.uk/uniprot/) (Bairoch and Apweiler, 2000). A whole genome Blast search (E-value less than 1e-5, minimal alignment length percentage larger than 40%) was performed against above seven databases.” in the revised manuscript (Lines 333-351).

Comment 28: Line 462. Accession list of other taxa in the supplement would help here.

Thanks for your comments. The accession numbers of these strains were displayed behind these strains in Figure 1A. According to your suggestions, we have added an accession list of these taxa (Supplementary Table 6) in the revised manuscript.

Comment 29: Line 463. Is there any literature to support that these are phylogenetically informative genes for Inoviridae?

Thanks for your comments. There are some literatures (Zeng et al, 2021; Evseev et al, 2023) to support that these are phylogenetically informative genes for *Inoviridae.* We have added these literatures in the revised manuscript.

References related to this response:

Zeng, J., Wang, Y., Zhang, J., Yang, S., and Zhang, W. (2021) Multiple novel filamentous phages detected in the cloacal swab samples of birds using viral metagenomics approach *Virol J* 18:240

Evseev, P., Bocharova, J., Shagin, D., and Chebotar, I. (2023) Analysis of *Pseudomonas aeruginosa* isolates from patients with cystic fibrosis revealed novel groups of filamentous bacteriophages. *Viruses* 15: 2215

**Reviewer #2 (Public Review):**
Summary: This paper investigates virus-host interactions in deep-sea bacteriophage systems which employ a seemingly mutualistic approach to viral replication in which the virus aids host cell polysaccharide import and utilization via metabolic reprogramming. The hypothesis being tested is supported with solid and convincing evidence and the findings are potentially generalizable with implications for our understanding of polysaccharide-mediated virus-host interactions and carbon cycles in marine ecosystems more broadly.

Thanks for your positive comments.

Strengths: This paper synthesizes sequencing and phylogenic analyses of two *Lentisphaerae* bacteria and three phage genomes; electron microscopy imaging of bacterial/phage particles; differential gene expression analyses; differential growth curve analyses, and differential phage proliferation assays to extract insights into whether laminarin and starch can induce both host growth and phage proliferation. The data presented convincingly demonstrate that both host culture density and phage proliferation increase as a result having host, phage, and polysaccharide carbon source together in culture.

Thanks for your positive comments.

Weaknesses (suggestions for improvement):(1) The article would be strengthened by the following additional experiment: providing the phage proteins hypothesized to be aiding host cell growth (red genes from Figure 5...TonB system energizer ExbB, glycosidases, etc) individually or in combination on plasmids rather than within the context of the actual phage itself to see if such additional genes are necessary and sufficient to realize the boosts in host cell growth/saturation levels observed in the presence of the phages tested.

Thanks for your valuable comments. It is a really good idea to express individually or in combination on plasmids to see the effects of those polysaccharide-degradation proteins in the host cell. However, at present, we failed to construct the genetic and expression system for the strictly anaerobic strain WC36, which hindering our further detailed investigation of the functions of those polysaccharide-degradation proteins. In our lab, we are trying our best to build the genetic and expression system for strain WC36. We will definitely test your idea in the future.

(2) The paper would also benefit from additional experiments focused on determining how the polysaccharide processing, transport, and metabolism genes are being used by the phages to either directly increase viral infection/replication or else to indirectly do so by supporting the growth of the host in a more mutualistic manner (i.e. by improving their ability to import, degrade, and metabolize polysaccharides).

Thanks for your valuable comments. Indeed, due to the chronic phage genome is not within the chromosome of the host, it is very hard to disclose the exact auxiliary process and mechanism of chronic phages. At present, we are trying to construct a genetic manipulation system for the strictly anaerobic host WC36, and we will gradually reveal this auxiliary mechanism in the future. In addition, combined with the reviewer 1’s suggestions, the focus of revised manuscript is to emphasize that polysaccharides induce deep-sea bacteria to release chronic phages, and most of the content of phage assisting host metabolism of polysaccharides has been deleted.

(3) The introduction would benefit from a discussion of what is known regarding phage and/or viral entry pathways that utilize carbohydrate anchors during host entry. The discussion could also be improved by linking the work presented to the concept of "selfishness" in bacterial systems (see for instance Giljan, G., Brown, S., Lloyd, C.C. et al. Selfish bacteria are active throughout the water column of the ocean. ISME COMMUN. 3, 11 (2023) https://doi.org/10.1038/s43705-023-00219-7). The bacteria under study are gram negative and it was recently demonstrated (https://www.nature.com/articles/ismej201726) that "selfish" bacteria sequester metabolizable polysaccharides in their periplasm to advantage. It is plausible that the phages may be hijacking this "selfishness" mechanism to improve infectivity and ENTRY rather than helping their hosts to grow and profilerate so they can reap the benefits of simply having more hosts to infect. The current work does not clearly distinguish between these two distinct mechanistic possibilities. The paper would be strengthened by at least a more detailed discussion of this possibility as well as the author's rationale for interpreting their data as they do to favor the "mutualistic" interpretation. In the same light, the paper would benefit from a more careful choice of words which can also help to make such a distinction more clear/evident/intentional. As currently written the authors seem to be actively avoiding giving insights wrt this question.

Thanks for your valuable comments. According to your suggestions, we have added the related discussion as “Moreover, it was recently demonstrated that selfish bacteria, which were common throughout the water column of the ocean, could bind, partially hydrolyze, and transport polysaccharides into the periplasmic space without loss of hydrolysis products (Reintjes et al., 2017; Giljan et al., 2023). Based on our results, we hypothesized that these chronic phages might also enter the host through this “selfishness” mechanism while assisting the host in metabolizing polysaccharides, thus not lysing the host. On the other hand, these chronic phages might hijack this “selfishness” mechanism to improve their infectivity and entry, rather than helping their hosts to grow and proliferate, so they could reap the benefits of simply having more hosts to infect. In the future, we need to construct a genetic operating system of the strictly anaerobic host strain WC36 to detailedly reveal the relationship between chronic phage and host.” in the revised manuscript (Lines 305-316).

References related to this response:

Reintjes, G., Arnosti, C., Fuchs, B.M., and Amann, R. (2017) An alternative polysaccharide uptake mechanism of marine bacteria *ISME J* 11:1640-1650

Giljan, G., Brown, S., Lloyd, C.C., Ghobrial, S., Amann, R., and Arnosti, C. (2023) Selfish bacteria are active throughout the water column of the ocean *ISME Commun* 3:11

(4) Finally, I would be interested to know if the author’s sequencing datasets might be used to inform the question raised above by using bacterial immunity systems such as CRISPR/Cas9. For example, if the phage systems studied are truly beneficial/mutualistic for the bacteria then it’s less likely that there would be evidence of targeted immunity against that particular phage that has the beneficial genes that support polysaccharide metabolism.

Thanks for your comments. According to your suggestions, we have carefully analyzed the genome of strain WC36, and found that there were no CRISPR/Cas9-related genes. Considering our results that the number of chronic phages was increased with the prolongation of culture time, we speculated that host might have no targeted immunity against these chronic phages.

**Reviewer #2 (Recommendations For The Authors):**
There are some minor grammatical errors and unclear statements (lines 99-100, 107-109, 163, 222, 223, 249-250, 254) which should also be fixed before final publication.

Thanks for your valuable comments. We have fixed these minor grammatical errors and unclear statements in the revised manuscript.

Lines 99-100: we have modified this description as “For instance, AMGs of marine bacteriophages have been predicted to be involved in photosynthesis (Mann et al., 2003), nitrogen cycling (Ahlgren et al., 2019; Gazitúa et al., 2021), sulfur cycling (Anantharaman et al., 2014; Roux et al., 2016), phosphorus cycling (Zeng and Chisholm, 2012), nucleotide metabolism (Sullivan et al., 2005; Dwivedi et al., 2013; Enav et al., 2014), and almost all central carbon metabolisms in host cells (Hurwitz et al., 2013).” in the revised manuscript (Lines 100-105).

Lines 107-109: we have modified this description as “However, due to the vast majority of deep-sea microbes cannot be cultivated in the laboratory, most bacteriophages could not be isolated.” in the revised manuscript (Lines 110-111).

Line 163: we have modified this description as “Based on the growth curve of strain WC36, we found that the growth rate of strictly anaerobic strain WC36 was relatively slow.” in the revised manuscript (Lines 149-151).

Lines 222-223: we have modified this description as “Regardless of whether the laminarin was present, the bacterial cells kept their cell shape intact, indicating they were still healthy after 30 days” in the revised manuscript (Lines 195-197).

Lines 249-250: we have modified this description as “However, the entry and exit of the hexagonal phages into the WC36 cells were not observed.” in the revised manuscript (Lines 190-191).

Line 254: we have modified this description as “To explore whether the production of bacteriophages induced by polysaccharide is an individual case, we further checked the effect of polysaccharides on another cultured deep-sea *Lentisphaerae* strain zth2.” in the revised manuscript (Lines 213-215).